



# From the Sun to the Earth: August 25, 2018 geomagnetic storm effects

Mirko Piersanti[1], Paola De Michelis[2], Dario Del Moro[3], Roberta Tozzi[2], Michael Pezzopane[2], Giuseppe Consolini[4], Maria Federica Marcucci[4], Monica Laurenza[4], Simone Di Matteo[5], Alessio Pignalberi[2], Virgilio Quattrociocchi[4,6], and Piero Diego[4]

[1]INFN - University of Rome "Tor Vergata", Rome, Italy.
[2]Istituto Nazionale di Geofisica e Vulcanologia, Rome, Italy.
[3]University of Rome "Tor Vergata", Rome, Italy.
[4]INAF-Istituto di Astrofisica e Planetologia Spaziali, Rome, Italy.
[5]Catholic University of America at NASA Goddard Space Flight Center, Greenbelt, Maryland, USA
[6]Dpt. of Physical and Chemical Sciences, University of L'Aquila, L'Aquila, Italy

**Correspondence:** Mirko Piersanti (mirko.piersanti@roma2.infn.it)

**Abstract.** On August 25, 2018 the interplanetary counterpart of the August 20, 2018 Coronal Mass Ejection (CME) hit the Earth, giving rise to a strong G3 geomagnetic storm. We present a description of the whole sequence of events from the Sun to the ground as well as a detailed analysis of the observed effects on the Earth's environment by using a multi instrumental approach. We studied the ICME propagation in the interplanetary space up to the analysis of its effects in the magnetosphere, ionosphere and at ground. To accomplish this task, we used ground and space collected data, including data from CSES (China Seismo Electric Satellite), launched on February 11, 2018. We found a direct connection between the ICME impact point onto the magnetopause and the pattern of the Earth's polar electrojects. Using the Tsyganenko TS04 model prevision, we were able to correctly identify the principal magnetospheric current system activating during the different phases of the geomagnetic storm. Moreover, we analyzed the space-weather effects associated with the August 25, 2018 solar event in terms of evaluation geomagnetically induced currents (GIC) and identification of possible GPS loss of lock. We found that, despite the strong geomagnetic storm, no loss of lock has been detected. On the contrary, the GIC hazard was found to be potentially more dangerous than other past, more powerful solar events, such as the St. Patrick geomagnetic storm, especially at latitudes higher than $60°$ in the European sector.

## 1 Introduction

Geomagnetic storms and substorms are among the most important signatures of the variability in the solar-terrestrial relationships. They are extremely complicated processes, which are triggered by the arrival of solar perturbations, such as coronal mass ejections (CMEs) (e.g. *Gosling*, 1993; *Bothmer and Schwenn*, 1995; *Gonzales and Tsurutani*, 1987; *Piersanti et al.*, 2017b)



and affect the entire magnetosphere. Indeed, these processes are both highly non linear and multiscale, involving a wide range

of plasma regions and phenomena in both the magnetosphere and ionosphere that mutually interact. Computer simulations, ground-based and space-borne observations, over the last thirty years, have highlighted such strong feedback and coupling processes (*Piersanti et al.*, 2017b, and reference therein). This is the reason why, to properly understand geomagnetic storms and magnetospheric substorms, it is necessary to consider the entire chain of the processes as a single entity.

When these processes are analyzed, one has always to consider that the dynamic pressure of the solar wind and the inter-

planetary magnetic field (IMF) control the strength and the spatial structure of the magnetosphere-ionosphere current systems, whose changes are at the origin of geomagnetic activity, i.e. of the variation of the Earth's magnetospheric-ionospheric field as observed by space and ground-based measurements. Indeed, a significant amount of the Sun energy can be dropped off either directly in the polar ionosphere or in the form of magnetic field energy in the equatorial central regions (the central plasma sheet, the current sheet, etc.) of the Earth's magnetospheric tail, from where it is successively injected into the inner magneto-

spheric regions such as, for instance, the radiation belts (*Gonzalez et al.*, 1994). The growth of the trapped particle population in the inner magnetosphere produces a significant increase of the ring current, while the energy released from the magnetotail and injected into the high latitude ionosphere, together with that directly deposited in the polar regions, is responsible for an enhancement of the auroral electrojet current systems (*McPherron*, 1995). The importance of studying these processes lies not only in understanding the physical processes which characterise the solar-terrestrial environment, but also in its impact on the

technological and anthropic systems. Indeed, nowadays geomagnetic storms and substorms have become an important concern, being potentially able to damage the anthropic infrastructures at ground and in space, as well as of harming human health (e.g. *Baker*, 2001; *Ginet*, 2001; *Kappenman*, 2001; *Lanzerotti*, 2001; *Pulkkinen et al.*, 2017). As a consequence, these processes play an important role in the space weather framework where the applications and societal relevance of the phenomena are much more explicit than in solar-terrestrial physics (*Koskinen et al.*, 2017).

In this paper, we analysed a recent solar event occurred on August 20, 2018, which affected the Earth's environment on August 25, 2018, giving rise to a G3 geomagnetic storm. We used a transversal approach to describe the whole sequence of events from the Sun to the ground. We carried out an interdisciplinary study that, starting from the analysis of the CME at the origin of the storm, of its propagation in the interplanetary space (hereafter, Interplanetary CME - ICME), down to the analysis of the effects produced by the arrival of this perturbation in the magnetosphere, ionosphere and at ground. We

used measurements recorded on board satellites and at ground stations, in order to both follow the event evolution and focus our attention on its ionospheric and geomagnetic effects measured at different latitudes/longitudes. Namely, we discuss how the activity of the solar atmosphere and solar wind, travelling in the interplanetary space, has been able to deeply influence the conditions of the Earth's magnetosphere and ionosphere or more generically has been able to deeply influence the solar-terrestrial environment. We studied the propagation through the heliosphere of the CME, trying to take into consideration

the complicated and multifaceted nature of its interaction with the ambient solar wind and the magnetosphere, and on the geomagnetic and ionospheric effects caused by this event. We exploit data from both satellites and ground-based observatories, whose integration is fundamental to describe the effects on the Earth's environment produced by solar activity. We collected and processed data from low-Earth orbit satellites, as for example ESA-Swarm and CSES (China Seismo Electromagnetic





Satellite), and from ground-based magnetometers. More than 80 magnetic observatories located all over the Globe (all those
available for the period under investigation), were involved in the analysis. To detect ionospheric irregularities, we used the
Rate Of change of electron Density Index (RODI) estimated from the electron density measured on board of CSES to study
the occurrence of electron density fluctuations. Finally, we evaluated possible geomagnetically induced current hazard related
to the main phase of the August 2018 geomagnetic storm, calculating the Geomagnetically Induced Current (GIC) index
(*Marshall et al.*, 2010; *Tozzi et al.*, 2019) over two geomagnetic quasi-longitudinal array located in the European-African and
in the North American sectors.

## 2   CME - Interplanetary propagation

The solar event that has been associated with the magnetospheric disturbances under analysis, occurred on August 20, 2018.
The source was an extremely slow CME that was not detected by SOHO LASCO (*Domingo et al.*, 2016; *Bothmer et al.*, 1995)
and would be therefore classified as a stealth CME (*Howard and Harrison*, 2013), if it was not imaged by STEREO-A COR2
(*Keiser et al.*, 2008; *Howard et al.*, 2013). While propagating, the ICME created an interplanetary shock that, at L1, advanced
the ICME itself by more than 30 hours. We note that the creation of a shock is not incompatible with a slow CME, since the
shock can be created by the expansion of the CME as it equalizes its pressure with the interplanetary plasma. In this section,
we present the characteristics of the CME at lift-off and of the ICME at L1, and put forward an interpretation of its propagation
by using a modified Drag Based Model (P-DBM, *Vrsnak et al.*, 2013; *Napoletano et al.*, 2018).

### 2.1   CME lift-off and interplanetary response

While the CME was hardly visible in the FoVs of SOHO LASCO instruments, it could be easily seen in STEREO-A COR2
images, with an angular width of $\simeq 45°$. The CME appears as a diffuse, slow plasma structure, entering COR2 FoV (Field of
View) on August 20, 2018 at 16:00 UT ($\pm 1$ hr) and reaching the FoV edge on August 21, 2018 at 08:00 UT ($\pm 2$ hr). From this
timing, we can estimate a PoS (Plane of Sky) velocity for the CME $V_{PoS} = (160 \pm 40)$ km/s.

The most probable source for the CME is a filament eruption observed on August 20, 2018 at $t_0$= 08:00 UT at heliographic
coordinates $\theta_{Sun} = 16°, \phi_{Sun} = 14°$ on the solar surface (Pink post in Figure 1). The filament ejection has been recorded by
SDO AIA (*Pesnell et al.*, 2019; *Lemen et al.*, 2011) imagers. Considering the relative positions of STEREO-A at the moment of
the CME lift-off, the source on the Sun, the information provided by the CDAW catalog of CME, and the hypothesis of radial
propagation, we can de-project the CME velocity and estimate its radial velocity at about 10 $R_{Sun}$ as $V_{rad} = (350 \pm 45)$ km/s.
In this respect, we report that the derived radial velocity is lower that the median of the CME speed distribution (*Yurchyshyn et
al.*, 2005), and confirms that CMEs associated with filament eruption tend to be slower than those associated with flares (e.g.,
*Moon et al.*, 2002).

We also note that, at the time of lift-off, a sizeable coronal hole (Yellow posts in Figure 1) was present at heliographic coordi-
nates $\theta_{Sun} \simeq -8°, \phi_{Sun} \simeq -20°$ that would generate a fast solar wind stream that could affect the CME propagation.

Figure 2 shows the ICME detection by Wind (*Lepping et al.*, 2019), DSCOVR (*Burt and Smith*, 2012), and ACE (*Stone et al.*,





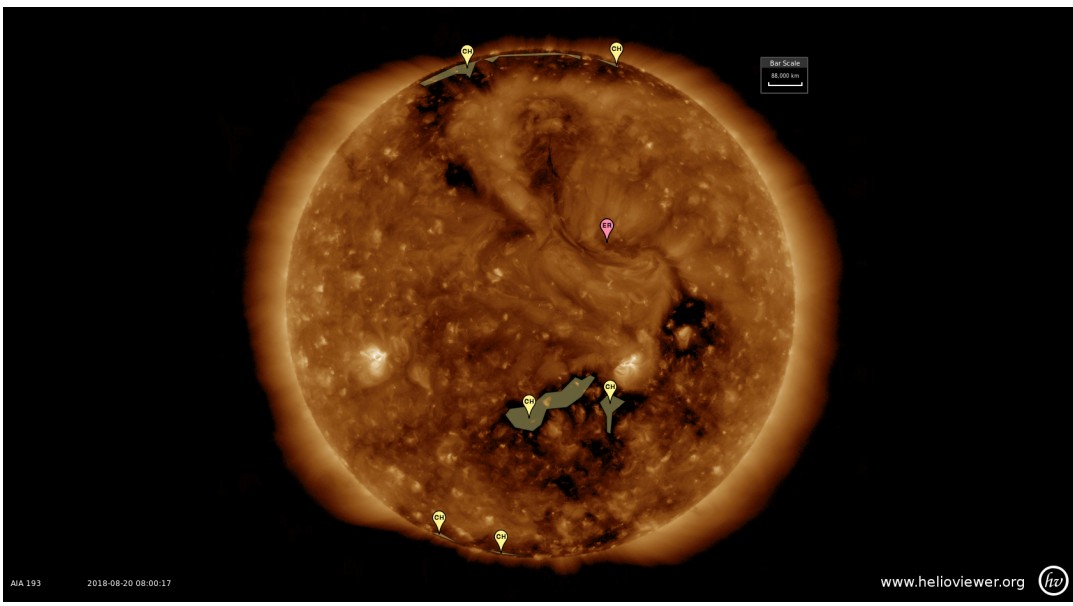

**Figure 1.** Image of the Sun with EUV SDO AIA193 at the time of the filament eruption. The pink post marks the position of the filament eruption associated with the CME; yellow posts and yellow shaded areas mark the position of coronal holes. Image created using the ESA and NASA funded Helioviewer Project.

2017) spacecrafts located at approximately L1 point. An interplanetary (IP) shock passed the three spacecrafts respectively at $\sim$05:37 UT, $\sim$05:42 UT, and $\sim$05:43 UT on August 24, 2018.

The IP shock was characterized by small variation of the solar wind (SW) density ($\Delta n_{p,W} \approx 2.5$ cm$^{-3}$, $\Delta n_{p,D} \approx 2.8$ cm$^{-3}$, $\Delta n_{p,A} \approx 1.8$ cm$^{-3}$), velocity ($\Delta v_{SW,W} \approx 18$ km/s, $\Delta v_{SW,D} \approx 16$ km/s, $\Delta v_{SW,A} \approx 16$ km/s), pressure ($\Delta P_{SW,W} \approx 0.9$ nPa, $\Delta P_{SW,D} \approx 0.9$ nPa, $\Delta P_{SW,A} \approx 0.7$ nPa), and interplanetary magnetic field (IMF) strength ($\Delta B_{IMF,W} \approx 0.8$ nT, $\Delta B_{IMF,D} \approx 1.1$ nT, $\Delta B_{IMF,ACE} \approx 1$ nT).

In agreement with the Rankine-Hugoniot conditions, the shock normal for the three spacecrafts was oriented at $\Theta_{SE,W} \approx -45°$ and $\Phi_{SE,W} \approx 130°$, $\Theta_{SE,D} \approx -45°$ and $\Phi_{SE,D} \approx 140°$, $\Theta_{SE,A} \approx -50°$ and $\Phi_{SE,A} \approx 100°$. The estimated shock speeds were respectively $v_{sh,W} \approx 300$ km/s, $v_{sh,D} \approx 300$ km/s, and $v_{sh,A} \approx 340$ km/s. Therefore, the predicted time of the impact of the IP shock onto the magnetosphere was at 06:14 UT (32 minutes after DSCOVR observations). The predicted location of the shock impact at the magnetopause, assuming a planar propagation, was at 7:00 ($\pm$00:15) LT (i.e. in the morning side of the magnetopause), corresponding, in the ecliptic plane, to $X_{GSE} = 5.0$ ($\pm 0.2$)$R_E$ and $Y_{GSE} = -20.0$ ($\pm 0.2$)$R_E$ (GSE is the Geocentric Solar Ecliptic reference system and $R_E$ is the Earth's radius) (Figure 2g).

The August 20 ICME included a significant magnetic cloud, observed at the Earth's orbit between August 25 at $\sim$12:15 UT and August 26 at $\approx$10:00 UT, whose boundaries are determined (*Burlaga et al.*, 1981) according to the the magnetic field behaviour conjoint with the temperature, the velocity and the density of protons, as depicted in Figure 2: the plasma temperature decreases from $\sim 9 \cdot 10^4$ K to $\sim 1.5 \cdot 10^4$ K; the total magnetic field increases to 16 nT, remaining there for approximately 12





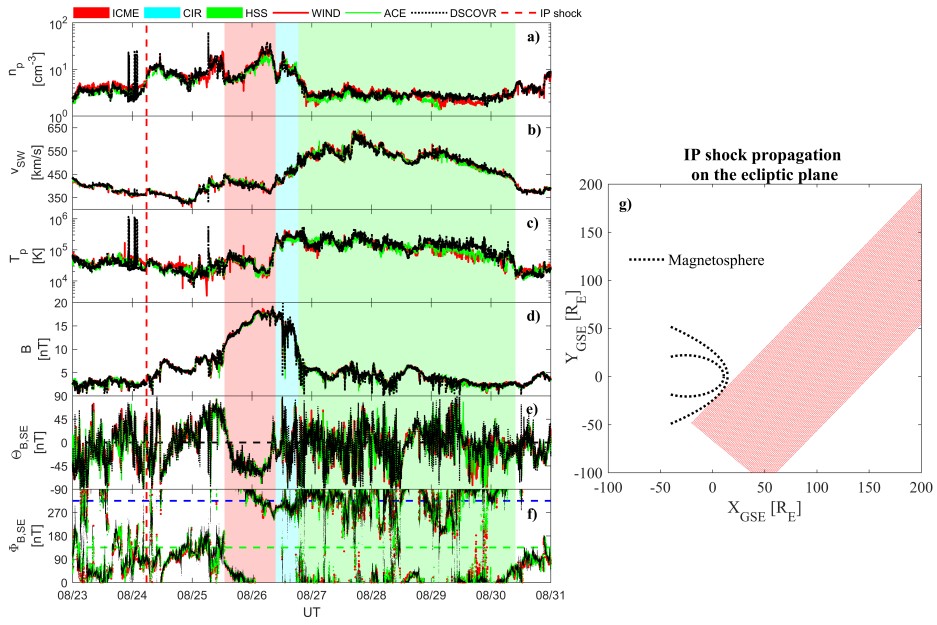

**Figure 2.** Solar wind parameters observed by WIND (red), ACE (green), and DSCOVR (black) spacecraft at L1: a) proton density; b) velocity; c) proton temperature; d) IMF intensity; e-f) IMF orientation ($\Theta_{SE}$ and $\Phi_{SE}$, respectively) in the SE coordinate system. The vertical horizontal green and blue lines in panel f represent the expected orientation of the Parker spiral at L1. The red dashed line indicates the interplanetary shock as observed on August 24 at $\approx$ 5:43 UT (to match ACE measurements, both WIND and DSCOVR data were translated of 6 min and 1 min, respectively). The red shaded region identifies the ICME. The cyan and green shaded regions shows the CIR and the HSS, respectively; g) Interplanetary shock propagation in the ecliptic plane.

hours; the magnetic field smoothly rotated, leading to pronounced and prolonged southward orientation (beginning at $\approx$14:30 UT on August 25) for approximately 22 hours; the solar wind speed fluctuated between $\sim$450 km/s and $\sim$370 km/s. A co-

rotating interaction region (CIR) followed on August 26, the solar wind plasma showing a velocity (temperature) increase at $\approx$10:00 UT from $\sim$370 km/s ($\sim 4 \cdot 10^4$ K) to near $\sim$550 km/s ($\sim 30 \cdot 10^4$ K) at $\approx$12:20 UT, a density decrease from $\sim 11 cm^{-3}$ to $\sim 3 cm^{-3}$, as the solar wind stream was transitioning into a negative polarity High Speed Stream (HSS).

## 2.2 A model for the propagation of the ICME

To describe the ICME propagation in the heliosphere we used the P-DBM (*Napoletano et al.*, 2018; *Del Moro et al.*, 2019)

model. Considering the presence of the Coronal Hole (CH) on the Sun at the time of the CME lift-off and the CIR observations in in-situ data, we proposed the following scenario, where:

- the ICME propagation is longitudinally deflected by its interaction with the solar wind, as in equation 8 of *Isavnin et al.* (2013);

- the ICME is later overtook by the fast solar wind stream from the identified CH at a distance $r_{Mix}$;




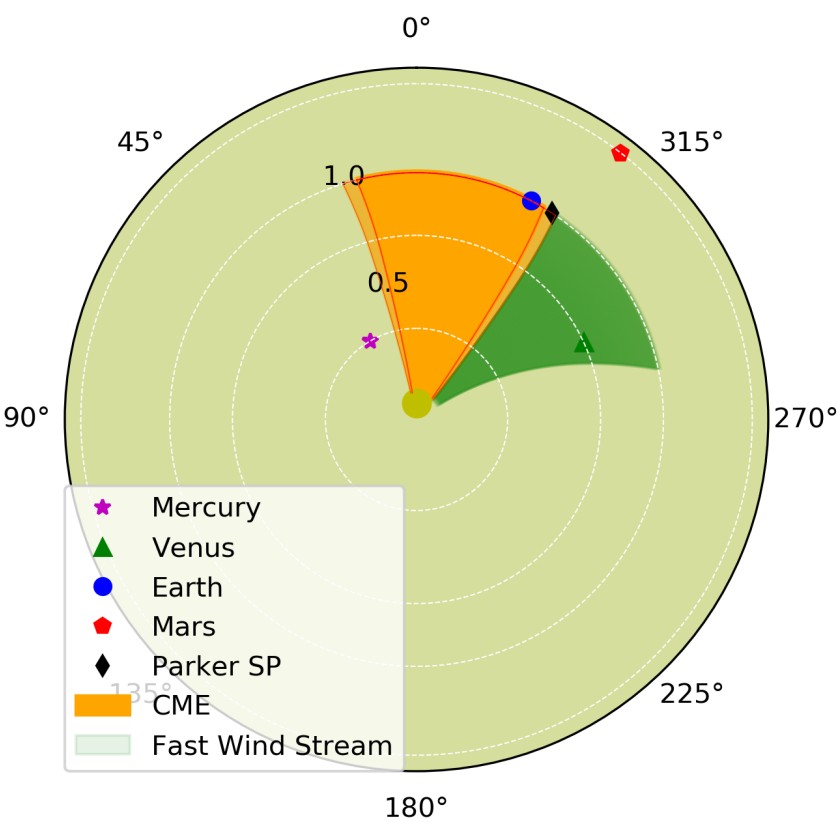

**Figure 3.** Scheme for the propagation of the CME in the inner heliosphere. The positions of the inner planets and Parker Solar Probe at the time of the ICME arrival at 1AU are represented by colored symbols. The ICME trajectory computed by the P-DBM is represented by the orange shadowed area. The lighter orange areas represent the $1\sigma$ uncertainty about the ICME trajectory from the 10000 different model runs. The green shadowed area represents instead the fast solar wind stream.

– $r_{Mix}$ is computed considering the time for the CH to rotate in the appropriate direction plus the time for the stream to catch up with the ICME.

Applying the same philosophy behind the P-DBM, the longitude of the fast wind stream, generated by the CH, has been associated with a $2.5°$ error with a Gaussian distribution.

From 10000 runs of this model, the most probable result are: the ICME arrival time and velocity at 1AU are August 25, 2018

at $t_{1AU}$= 16:00 UT ($\pm 9$ hr) and $V_{1AU} = 440$ ($\pm 70$) km/s, respectively; the fast solar wind stream interacts with the ICME beyond $r_{Mix} = 1.1$ ($\pm 0.1$) AU. These values agree nicely with the ICME actual arrival characteristics estimates as derived in the previous section.





As discussed in *Richardson* (2018), a CIR would form by the interaction of a HSS with the preceding slower (in this case)
ICME. Approximately one day later than $t_{1AU}$, the rotation of the Sun brings the CIR to sweep over Earth position, followed

by a HSS. Last, this model predicts that the ICME that hit Earth, would instead miss Mars and quite probably also the then
newly launched Parker Solar Probe (PSP - *Fox et al.*, 2016). While no data is available for the PSP at that date, no solar particle
event was actually detected in the following days by the instrumentation on-board MAVEN (*Jakosky et al.*, 2015).

A graphical representation of this result is shown in Fig. 3, where the position of the inner planets and of the Parker Solar Probe
at $t_{1AU}$ are represented by colored symbols. The orange area represents the trajectory of the ICME, with lighter orange areas

representing the $1\sigma$ uncertainty about its trajectory from the 10000 different model runs. The green area represents the part of
the inner heliosphere affected by the HSS at $t_{1AU}$.

## 3 Magnetospheric-Ionospheric system response

A complete and accurate knowledge of the magnetospheric-ionospheric coupling and of its dynamics in response to the changes
of the interplanetary medium conditions, is critical to many aspects of the space weather.It is, indeed, well-known that the

changes of the IMF and of the solar wind features, in terms of magnetic field orientation, plasma density, velocity, etc., is
capable of generating a fast increasing of the magnetospheric-ionospheric current intensities which manifests in multiscale
and rapid fluctuations of ground-based magnetic field. The response of the magnetosphere-ionosphere system to interplanetary
changes is however the consequence of both directly-driven, i.e., large scale plasma convection enhancement, and triggered-
internal phenomena, such as loading-unloading mechanisms, sporadic plasma energizations in the magnetotail, bursty-bulk

flows (*Milan*, 2017). The response of such a system is strongly dependent on the magnetospheric plasma internal state, with
a specific emphasis to the magnetotail central plasma sheet status. The result of the interplay between internal dynamics
and directly-driven processes is a very complex dynamics showing scale invariant features typical of non-equilibrium critical
phenomena (*Consolini et al.*, 1996; *Consolini*, 1997; *Consolini and De Michelis.*, 1998; *Consolini*, 2002; *Lui et al.*, 2000; *Sitnov
et al.*, 2001; *Uritsky and Pudovkin*, 1998; *Uritsky et al.*, 2002). In a series of recent papers (*Alberti et al.*, 2017, 2018; *Consolini*

*et al.*, 2018) it has been clearly shown the existence of a separation of timescales between directly-driven and triggered internal
timescales in the response of the Earth's magnetosphere-ionosphere current systems as estimated by means of geomagnetic
indices in the course of magnetic storms and substorms. This separation of timescales is one of the fingerprints of the complex
character of the geomagnetic response, which makes very difficult to get a reliable forecast of its short timescale dynamics.

In this section, we investigate the magnetospheric-ionospheric response during the August 2018 geomagnetic storm. On

one hand, the magnetosphere accumulates energy from the solar wind and dissipates it through geomagnetic storms, driving
large electrical currents. On the other hand, these currents close down into the ionosphere, producing large scale magnetic
disturbances, such as the auroral electrojects, the DP-2 current system, prompt penetrating electric field and so on (*Piersanti
et al.*, 2017b; *Pezzopane et al.*, 2019, and references therein). Some of these features and phenomena will be discussed in the
next sections for the investigated August 2018 geomagnetic storm.



## 3.1 Magnetosphere

Figure 4 a) shows the response of the magnetosphere to the IP2 arrival. According to the *Shue et al.* (1998) model, the magnetopause nose moves inward up to $\sim$7.1 $R_E$. Indeed, the shape of the magnetospheric field lines before (black lines) and after (red lines) the shock impact, evaluated by means of the TS04 model(*Tsyganenko and Sitnov*, 2005) shows a large field compression. Correspondingly, GOES 14 (panels b, d and f) and GOES 15 (panels c, e and g) show, on August 25 at $\sim$ 06 : 30 UT, a strong compression ($\Delta B_{z,G14} = 10$ nT and $\Delta B_{z,G15} = 22$ nT) of the magnetic field coupled with a stretching of the magnetotail field lines, due to the northward IMF orientation (*Villante and Piersanti*, 2011; *Piersanti and Villante*, 2016; *Piersanti et al.*, 2017b, as already found by). This situation completely changes between August 25 at 13 : 55 UT and August 26 at 10 : 25 UT, corresponding to the arrival of the magnetic cloud. In fact, both GOES 14 and GOES 15 show a strong decrease of $B_z$ (panels f and g), interpreted in terms of magnetic reconnection between the magnetospheric field and the strong southward IMF ($\sim -20$ nT) observed in the corresponding interval (*Piersanti et al.*, 2017b, and references therein). Interestingly, both GOES satellites show a huge increase of the $B_x$ component (panels b and c) and a negative, then positive variation in the $B_y$ component (panels d and e). This behaviour is the signature of a strong stretching and twisting of the magnetospheric field lines during the main phase of the geomagnetic storm (*Piersanti et al.*, 2012, 2017b). This scenario is confirmed by a modified *Tsyganenko and Sitnov* (TS04* 2005) model indicated by red dashed lines in Figure 4. Model changes include: the magnetopause and the ring current alone, during the main phase; the concurring contribution of both the ring and the tail currents, during the recovery phase. TS04* model represents very well the magnetospheric observations at geosynchronous orbit, with an average correlation coefficient ($r$) for the three magnetic field components: $r =$0.92 for GOES14; $r =$0.75 for GOES15.

Figure 5 (box A) shows the CSES (China Seismo Electromagnetic Satellite) satellite (*Shen et al.*, 2017) magnetic observations (*Zhou et al.*, 2019) along North-South ($B_N$ - left panel), East-West ($B_E$ - central panel) and Vertical ($B_C$ - right panel) components after removing the internal and crustal contributions to the Earth's magnetic field (using the CHAOS-6 model *Finlay et al.*, 2016).

CSES is a Chinese satellite launched on February 11, 2018 hosting, among others, a fluxgate magnetometer, an absolute scalar magnetometer, two Langmuir probes, and two particle detectors. The satellite orbits at about 500 km of altitude (Low Earth Orbit - LEO) in a quasi-polar Sun-synchronous orbit and passes at about 14 and 2 local time (LT) in its ascending and descending orbits, respectively.

As expected (*Villante and Piersanti*, 2011), the greatest variations are observed along the horizontal components, where both the magnetospheric and ionospheric currents play a key role.

In order to quantify both the magnetospheric and ionospheric origin contributions at CSES orbit, we applied the MA.I.GIC. model (*Piersanti and Carter*, 2019) to discriminate between different time scales contributions in a time series. The results obtained are shown in Figure 5 (box B). Upper and lower panels report high ($\sim$25 $\mu$Hz $<$ f $<\sim$ 3 mHz; f being the frequency) and low frequency ($\sim$ 2.3 $\mu$Hz $<$ f $<\sim$ 25 $\mu$Hz) components observations, respectively. The low frequency behaviour shows a strong and rapid decrease along the North-South direction during the main phase of the geomagnetic storm and a long lasting increase during the recovery phase. On the other hand, $B_{E,LF}$ shows a negative then positive variation during the main and the



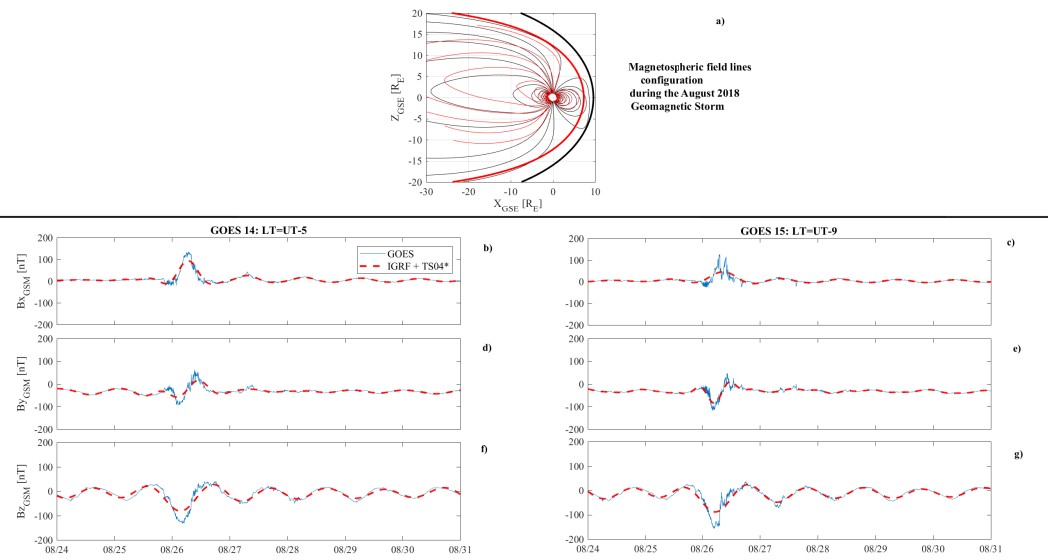

**Figure 4.** panel a): Magnetospheric field lines configurations as predicted by TS04 model before (black lines) and after (red lines) the IP shock; panels b), d) and f): magnetospheric field observations along $X_{GSM}$ (b), $Y_{GSM}$ (d) and $Z_{GSM}$ (f) at GOES 14 (LT=UT-5) geosynchronous orbit; panels c), e) and f): magnetospheric field observations along $X_{GSM}$ (b), $Y_{GSM}$ (d) and $Z_{GSM}$ (f) at GOES 15 (LT=UT-5) geosynchronous orbit; red dashed lines represent the IGRF+TS04* model prevision.

recovery phase, respectively. $B_{C,LF}$ is characterized by negligible variations. This behaviour is consistent with magnetospheric

origin field variations induced by the action of both the symmetric part of the ring current and tail current along $B_{N,LF}$ and of the asymmetric part of the ring current along $B_{E,LF}$ (*Piersanti et al.*, 2017b). It is confirmed by the comparison between the CSES magnetospheric origin contribution and the TS04* model (red lines in Figure 5, box B), in which we considered both the magnetopause and ring current alone during the main phase, and both the ring current and tail current alone during the recovery phase. It can be easily seen TS04* well represents the along $B_{N,LF}$ variations, while it is not able to reproduce the $B_{E,LF}$

variations. This would suggest that the partial ring current field (with the effect of the field-aligned currents associated with the local time asymmetry of the azimuthal near-equatorial current), which is not included in the TS04 model, plays a relevant role.

The high frequency components show large variations along both $B_{N,HF}$ and $B_{E,HF}$. This behaviour is consistent with the contributions due to the variations of the ionospheric current systems and to the magnetospheric-ionospheric coupling processes. In fact, the huge positive then negative variations observed during the main phase along both the horizontal components

can be imputable to the loading-unloading process between the magnetosphere and the ionosphere (*Consolini and De Michelis*, 2005; *Piersanti et al.*, 2017b). On the other hand, the variations observed during the recovery phase, which are positive on average, can be due to the ionospheric DP-2 current system (*Villante and Piersanti*, 2011; *Piersanti and Villante*, 2016; *Piersanti et al.*, 2017b).



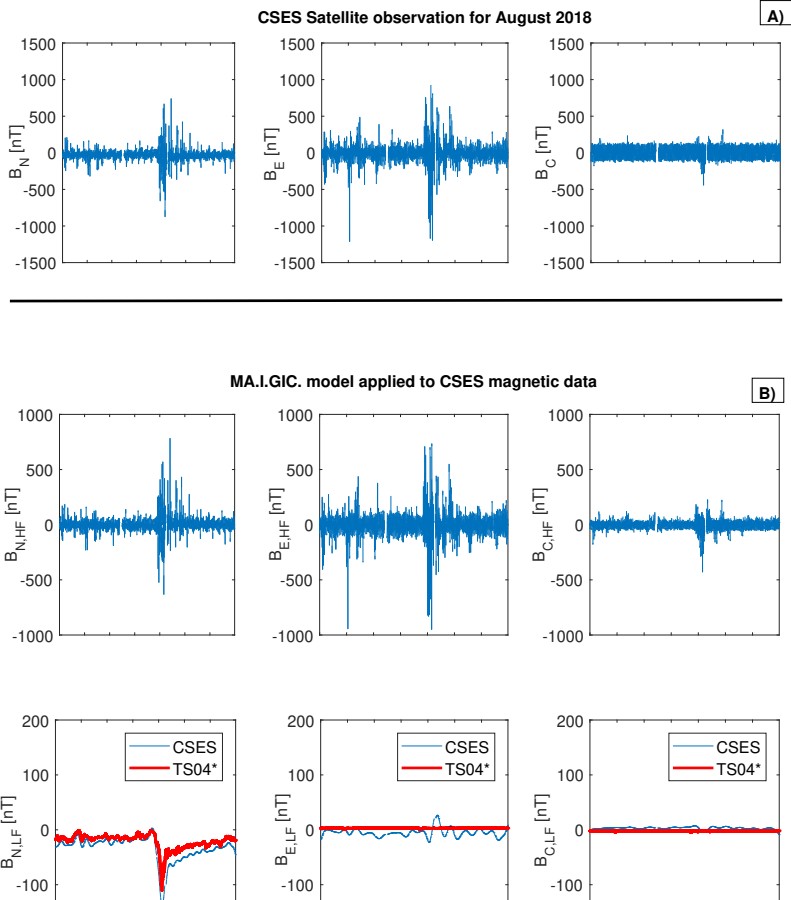

**Figure 5.** Box A) Magnetic field observations at CSES orbit along geographic North-South (left panel), East-West (middle panel) and vertical (right panel); Box B) MA.I.GIC. model applied to CSES magnetic data: upper panels show the high frequency time scales ($\sim 25\ \mu$Hz $<$ f $< \sim 3$ mHz; f being the frequency) for the three components of the observed field; lower panels show the low frequency time scales ($\sim 2.3$ $\mu$Hz $<$ f $< \sim 25\ \mu$Hz) for the three components of the observed field. Red lines represent the TS04* model previsions along CSES orbit.





## 3.2 Ionospheric response

The ionospheric plasma is often characterized by irregularities and fluctuations in the plasma density, especially during active solar conditions. In order to characterize such irregularities, we evaluated the RODI, a parameter derived from the electron density (see Appendix A) recorded by the CSES satellite (*Wang et al.*, 2019).

Figure 6 shows RODI values for August 25, 26, and 27, 2018, in which nighttime semi-orbits (around 02:00 LT) are shown separately from daytime semi-orbits (around 14:00 LT).

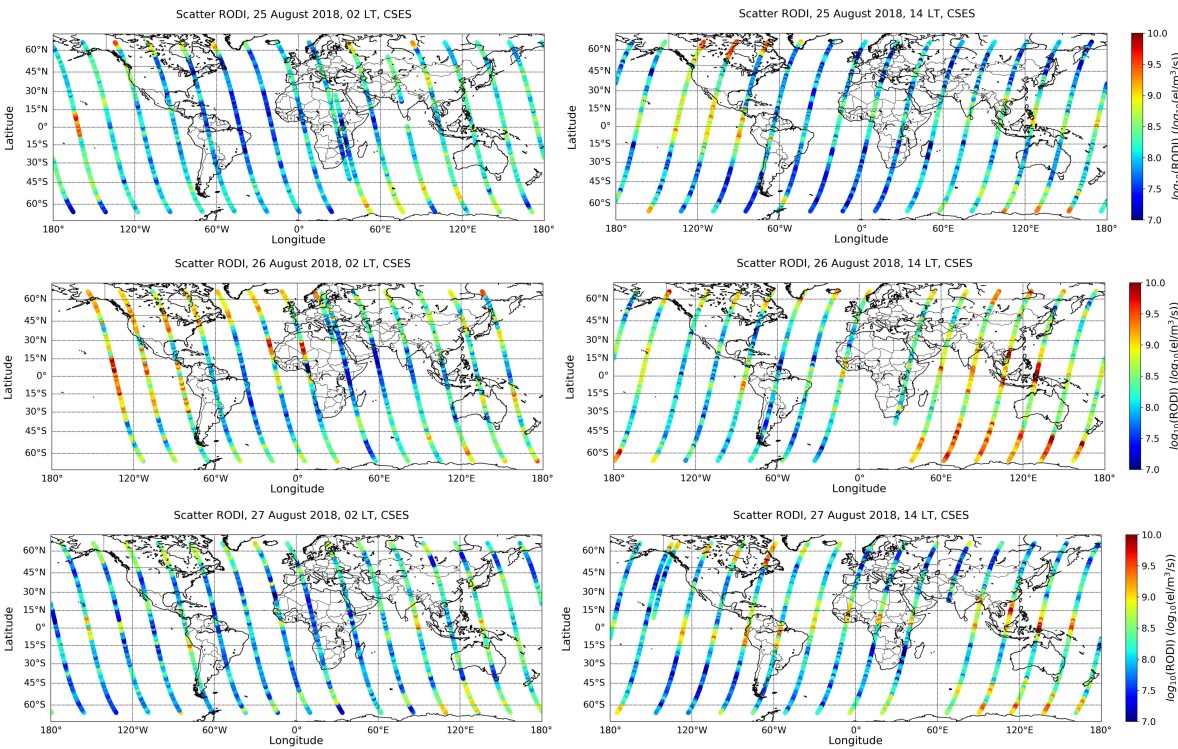

**Figure 6.** RODI calculated for August 25, 26, and 27, 2018. Scale is logarithmic. Coordinates are geographical. Left panels show nighttime semiorbits (ascending), while right panels show daytime semiorbits (descending). Time increases leftward. Electron density values are recorded by CSES.

A significant high RODI values, spreading all over the meridian during the main phase of the storm (August 25 and 26, 2018, especially the latter), for both nighttime and daytime, is clearly seen, while on August 27, 2018, the RODI comes back to lower values. This behaviour can be explained in terms of the presence, during the main phase, of ionospheric irregularities, especially at auroral and low latitudes. To understand whether this significant increase of irregularities could have caused space weather effects on navigational systems, we have considered vertical total electron content (vTEC) data measured by

Swarm satellites (*Friis-Christensen et al.*, 2006, 2008) to look for some loss of lock on GPS (Global Positioning System *Jin and Oksavik*, 2018, and references therein). Only vTEC data with corresponding elevation angles $\geq 50°$ have been taken into





account. We have considered vTEC data recorded by each of the three satellites (A, B, and C) of the Swarm constellation and corresponding to each PRN (Pseudo Random Noise) satellite in view. No loss of lock has been found, contrary to what happened, for instance, during the well-known and much more intense St. Patrick storm occurred on March 17, 2015 (*Jin and*

*Oksavik*, 2018; *De Michelis et al.*, 2016; *Pignalberi et al.*, 2016), where vTEC measurements highlighted many loss of lock (figures not shown). The fact that no loss of lock has been found during the August geomagnetic storm means that the event was weak in terms of space weather effects on navigational systems.

This fact is also supported by Figure 7, where rate of change of TEC index (ROTI) values (ROTI is calculated as RODI but considering TEC values in place of electron density values, for a defined GPS satellite in view) from Swarm A are shown for

PRN 8 on August 26, 2018, and for PRN 15 on March 17, 2015. It is clear, from this figure, that a loss of lock occurs when ROTI saturates, a feature that rarely happens on August 26, 2018 and more in general during the entire period under analysis.

## 4 Magnetic effects at ground

Space weather predictions and geomagnetic storms intensities are normally measured on the basis of well known geomagnetic indices. Anyway, as these indices are evaluated using ground observations (typically via magnetometers), it is crucial to

improve the knowledge of the effect of each magnetospheric and ionospheric current at ground. In this section, we focused on the ground magnetic response in terms of magnetospheric and ionospheric currents, and on the effects that those currents generated on the Earth's surface. GICs are one of the main ground effects of space weather events driven by solar activity (*Pulkkinen*, 2015; *Pulkkinen et al.*, 2017; *Carter et al.*, 2016; *Piersanti and Carter*, 2019). Since GICs represent the end of the space weather chain extending from the Sun to the Earth's surface, to complete the description of August, 25 2018 geo-

magnetic storm, an estimation of the amplitude of geomagnetically induced currents and of the associated risk level, to which power grids have been exposed during this storm, is also presented.

### 4.1 Geomagnetic field response

To analyse the magnetic effects at ground during the geomagnetic storm, we selected 83 magnetic observatories from IN-TERMAGNET magnetometer array network. INTERMAGNET is a consortium of observatories and operating institutes that

guarantees a common standard of data released to the scientific community, thus making it possible to compare the measurements carried out at different observation points. The distribution of the selected observatories is reported in Figure 8 and covers the geographic latitudes between $-80°$ and $80°$ providing a continuous sampling of the geomagnetic field. According to the standard of INTERMAGNET, the data consist of time series of the geomagnetic vector sampled at $\approx 1$ minute and filtered to avoid aliasing effects. We have considered the horizontal magnetic field component (H) and focused our analysis

on a period of seven days (from August 23 to August 29), during which the storm occurred. The selected period allows us to follow the evolution of the magnetic disturbance recorded at ground, during the geomagnetic storm. Moreover, we use the the model of (*Thomas and Shepherd*, 2018) based on the Super Dual Auroral Radar Network (SuperDARN) observations and the SuperDARN observations as well, to analyses the ionospheric convection during the same period. SuperDARN is an interna-



**Figure 7.** ROTI values from Swarm A calculated (top panel) for PRN 8 on August 26, 2018 and (bottom panel) for PRN 17 on March 17, 2015. Loss of lock visible in the figure, highlighted by blue circles, correspond to parts of the trace where ROTI saturates.

tional network of more than 35 high-frequency (HF) radars which has been implemented for the study of the ionosphere and
upper atmosphere at sub-auroral, auroral and polar cap latitudes in both Northern and Southern Hemispheres (*Chisham et al.*,
2007; *Nishitani et al.*, 2019).



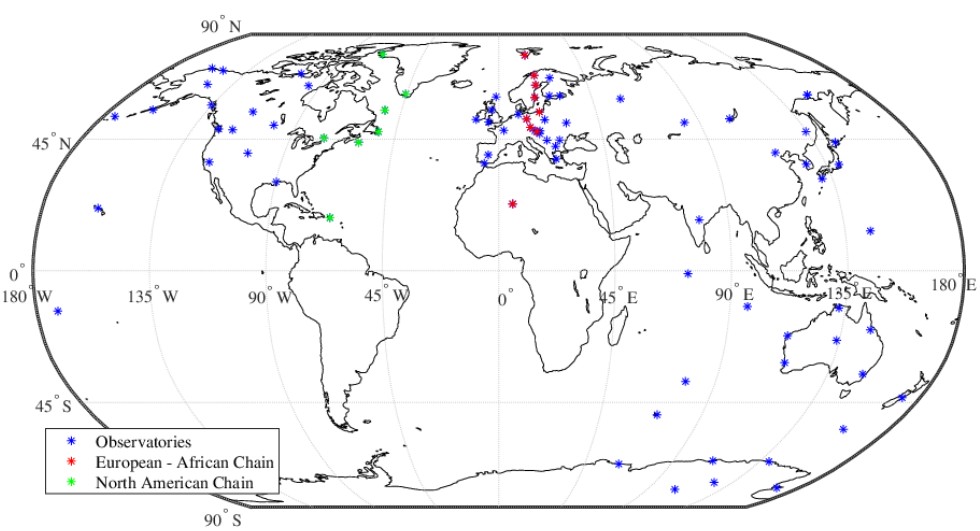

**Figure 8.** Geographical positions of the selected 83 INTERMAGNET geomagnetic observatories (blue stars). Red and green stars identifies a European-African and North American almost longitudinal chains, respectively, selected for the GIC analysis.

Figure 9 shows the daily distributions of the intensity of the horizontal magnetic field component obtained considering data recorded simultaneously by the selected magnetic observatories during the analysed period. The figure reports: on the left the values of the SYM-H index (*Menvielle*, 2011), which can be used to monitor the geomagnetic activity and more in detail the

ring current intensity during the geomagnetic storm; in the middle, daily polar view maps of the horizontal field magnitude in the Northern Hemisphere and of the ionospheric convection patterns derived from the model of (*Thomas and Shepherd*, 2018) based on SuperDARN observations; on the right, the cylindrical projection view of the same magnetic field component. Data are reported in geomagnetic latitude and magnetic local time (MLT, *Baker*, 1989).

Of particular interest is the analysis of the effects of the ionospheric and magnetospheric currents on the geomagnetic field.

For this reason, we have removed the main field from the data and considered only the magnetic fields generated by the electric currents in the ionosphere and magnetosphere (i.e. the so-called magnetic field of external origin). Thus, the horizontal field magnitude values reported in Figure 9 describe the magnetic field perturbations at ground due to external sources. The main contributions to this external field, producing relevant signatures in magnetic field observations, are the polar ionospheric


currents, such as the polar electrojets, and the magnetospheric currents such as the Chapman-Ferraro currents and (in particular)
the magnetospheric ring current (*Rishbeth and Garriot*, 1993; *Hargreaves*, 1992). These current systems are almost always
present even during geomagnetic quiet periods but show a significant variability during the disturbed periods (*De Michelis et al.*, 1997). The maps reported in Figure 9 show the effect due to the eastward and westward electrojets. These two polar current
systems, which are the most prominent currents at auroral latitudes, produce at ground a magnetic field perturbation that is
characterized by a positive excursion of the horizontal field magnitude in the case of the eastward electrojet, flowing in the
afternoon sector, and a negative one in the case of the westward electrojet flowing through the morning and midnight sector
(*De Michelis et al.*, 1999). It can be especially seen from the data reported in the polar view maps (central column in Figure
9). We noticed that these currents are always present but their intensities increase during the main phase of the geomagnetic
storm (*Ganushkina et al.*, 2018). Even their spatial distribution changes. Indeed, the magnetic disturbance, associated with
these electric currents, tends to shift towards lower latitudinal values drastically during the geomagnetic storm. on August 26
the westward electrojet is extremely intense and around midnight the effect at ground due to the substorm electrojet current
is recognizable, too. The associated disturbance fields cover the geomagnetic latitudes from $50°$ to $75°$ on the nightside.
Looking at the ionospheric convection as derived from the statistical model of (*Thomas and Shepherd*, 2018) and considering
that mean daily values of the IMF and solar wind velocity have been used as input to the model, the convection patterns
match the expansion to lower latitudes observed in the magnetic disturbance evolution during the extreme driving conditions
($E_{SW} \geq 4.0mV/m$) that characterise the period under study after the southward rotation of the IMF. In fact, also the convection
maps computed from the SuperDARN measurements at 2 minutes resolution (not shown) display that the auroral convection
zone expands equatorward to $50°$ geomagnetic latitude during the geomagnetic storm.The expansion of the convection pattern
is related to the dayside reconnection forming new open field lines once the IMF turned southward in late August 25.

The panels on the right column of Figure 9 show the effect due to the ring current that is responsible for a decrease of the
magnetic field intensity at low and mid latitudes, during the development of the geomagnetic storm. As known, the intensity of
the ring current increases during the main phase of a geomagnetic storm, because of the injection of energetic particles from
the magnetotail in the equatorial plane, and gradually decays during the recovery phase. The time evolution of the ring current,
through the time evolution of its associated disturbance field, is clearly visible in our data. During the main phase of the storm
(August 26), the increasing of the ring current flowing in the westward direction produces a strong depression of the horizontal
field magnitude, as can be seen by the blue region at mid/low-latitudes of the map corresponding to August 26, on the right-side
of Figure 9. In the days following the main phase the magnetic field perturbation associated with the ring current is still visible
at low and mid latitudes, although its amplitude rapidly decreases. We can conclude that the magnetic field perturbations on
the ground due to the arrival of the solar perturbation are clearly recognizable in the recorded data and are well in agreement
with what is expected from a theoretical point of view (*Piersanti et al.*, 2017b, and references therein).

**4.2 Ground magnetic effects**

Fluctuations of the geomagnetic field happening during geomagnetic storms or substorms are responsible for an induced
geoelectric field at the Earth's surface that, on turn, originates GIC that may represent an hazard for the secure and safe



operation of electrical power grids and oil/gas pipelines. For instance, for the case of power transmissions, GICs represent a hazard due to their frequency. Indeed, the power spectrum of the originating geoelectric field is dominated by frequencies smaller than 1 Hz and this makes of GIC a DC current that flows into 50-60 Hz AC power systems, with the consequence to temporarily or permanently damage power transformers (*Pulkkinen et al.*, 2017, and references therein).

As a proxy of the geoelectric field, and hence of GIC intensity, the GIC index (*Marshall et al.*, 2010) is calculated using the approach proposed by (*Tozzi et al.*, 2019). Among the proxies of the geoelectric field resorting to magnetic data only, this index has two main advantages: 1) it represents the geoelectric field better than other commonly used quantities (i.e. dB/dt or other geomagnetic activity indices), 2) its values are used to determine the risk level to which power networks are exposed during space weather events (*Marshall et al.*, 2011). Since the components of the geomagnetic field relevant for the induction of the geoelectric field are the horizontal ones, i.e. the Northward ($X$) and Eastward ($Y$) components, the GIC index is calculated for both of them. Particularly, $GIC_y$ and $GIC_x$ indices, are obtained using 1-minute of $X$ and $Y$ components, respectively, as observed at the geomagnetic observatories aligned along two latitudinal chains crossing North America and Europe-Africa. These two sets of observatories satisfy the condition to be characterised by geomagnetic longitudes in a range of about $\pm 20°$ around a central longitude. In the case of the North American chain the central geomagnetic longitude is about 17° E and the observatories used for this chain, indicated by their IAGA codes (refer to http://www.wdc.bgs.ac.uk/catalog/obs_code.html for observatories details) and ordered from high to low geomagnetic latitude, are: THL, NAQ, STJ, OTT, SBL, SJG, KOU. The central geomagnetic longitude of the European-African chain is about 105° E and the corresponding observatories, listed as above, are: HRN, ABK, LYC, UPS, HLP, NGK, BDV, TAM. To have an idea of the maximum GIC intensity produced by the August 26, 2018 geomagnetic storm, we calculated $GIC_x$ and $GIC_y$ indices for the geomagnetic observatories of the two chains and then picked out the maximum values reached by both GIC indices from August 25, 2018 at 18:00 UT to August 26, 2018 at 18:00 UT (i.e. the most geomagnetically disturbed conditions) and plotted them as a function of geomagnetic latitude in Figure 10. The two curves displayed in both panels a) and b) of Figure 10 refer to the North American (red) and to the European-African (blue) observatories chains, respectively. As expected, the latitudinal dependence of the maximum GIC intensity shows an increase with increasing latitude with a steepening of the curve around 60° N and then a substantial decrease at the highest latitude, near the geomagnetic pole. This reflects the geometry and the features of the current systems responsible for time variations of the geomagnetic field originating the induced geoelectric field. High-latitudes are affected by the effects of the auroral electrojets whose intensity undergo dramatic variations even increasing up to 4–5 times its quiet time value (*Smith et al.*, 2017). Low and mid latitudes are mainly affected by the ring current that produces variations of the geomagnetic field that are less effective for GICs building up. So, the peaks around 65-75° N, well visible in Figure 10, can be interpreted in terms of the position of the auroral oval and hence of the auroral electrojets flowing. Moreover, as can be observed by Figure 10 both the European-African and North American chains provide peaks of the GIC indices at different geomagnetic latitudes. In detail, the peak along the European-African chain seem to occur at latitudes smaller than that along the North American chain. Such observation can be explained in terms of the MLT at which the maxima of the GIC indices occur at the observatories of the two chains: around $(07:00 \pm 01:00)$ MLT for the European-African chain and around $(04:00 \pm 01:00)$ MLT for the North American chain. Indeed, as can be deduced by Figure 9, the maximum variation of the horizontal





component of the geomagnetic field recorded around 07:00 MLT occurs at latitudes lower than that observed at 04:00 MLT. The most the auroral oval expands towards lower latitudes, the smallest the latitude where the steepening of the maximum GIC

index occurs. Since, as already mentioned, the advantage to use the GIC index relates to the availability of an associated risk level scale, Figure 10 also displays colored dashed lines that indicate the boundaries between adjacent risk levels. This risk level scale has been introduced and defined by *Marshall et al.* (2011), it consists of four risk levels going from "very low" to "extreme", each associated with defined ranges of the $GIC_x$ and $GIC_y$ indices. This scale is based on a large occurrences of faults or failures of worldwide power grids and represents a probabilistic description of the threat, the risk level providing the

probability to have a fault; detailed information on this scale are given in *Marshall et al.* (2011). Results shown in Figure 10 tell that, for the analysed geomagnetic storm and for the same latitudes, power networks located along the European-African chain have been exposed to higher risk levels than those located along the North American chain.

As in the case of the ionospheric response, we repeated the analysis (same method and observatories), using data recorded during the 2015 St. Patrick geomagnetic storm (Figure 11), to have a quantitative comparison of the effects of the two storms.

There are evident similarities between Figure 11 and Figure 10, but some interesting differences can be highlighted. First, although 2015 St. Patrick storm has been slightly more intense than August 26, 2018 geomagnetic storm (minimum values of Sym-H index of -234 and - 206 nT, respectively), its maximum value of the GIC index is lower. Second, during St. Patrick's storm the southern boundary of the auroral oval has experienced a larger equatorward expansion. This can be deduced by the value of the southernmost latitudes exposed to risk levels higher than "moderate". In the case of the August storm, these are

larger than around 60° N, while during the St. Patrick storm they decreased to around 45-50° N. Last, the maximum values of GIC index at low-mid latitudes are very low for both geomagnetic storms but slightly higher in the case of St. Patrick storm. This suggests a greater participation of other current systems as, for instance, the ring current.

## 5  Summary and discussion

The solar event that has been associated with the August 25, 2018 geomagnetic storm occurred on August 20, 2018. The most

probable source for the CME is a filament eruption observed at 08:00 at heliographic coordinates $\theta_{Sun} = 16°, \phi_{Sun} = 14°$ on the solar surface (Pink post in Figure 1). The filament ejection has been recorded by SDO EUV imagers.

In order to reconstruct the ICME behavior in interplanetary space and to link the results from remote-sensing and in-situ data, we propagate the CME in the heliosphere in the framework of the P-DBM (*Napoletano et al.*, 2018) model under the hypotheses that: the ICME propagation is longitudinally deflected by its interaction with the solar wind; the ICME is later

overtook by fast solar wind stream from the identified coronal hole at a distance $r_{mix}$, evaluated considering the concurring contribution of both the time for the CH to rotate in the appropriate direction and the time for the stream to catch up with the ICME. It results that the ICME arrival time and velocity at 1AU are: August 25, 2018 at 16:00 UT ($\pm 9$ hr) and (440 $\pm 70$) km/s. This scenario is confirmed by the solar wind observations at L1. In fact, ACE, WIND and DSCOVR satellites detected the ICME arrival on August 25, 2018 at ~12:15 UT. The orientation of the IP shock normal preceding the ICME,

evaluated using the Rankine-Hugoniot conditions, was $\Theta_{SE} \approx -49°$ and $\Phi_{SE} \approx 135°$, with an estimated shock speeds of





$v_{sh} \approx 310$ km/s. As a consequence, the predicted time of the impact of the IP shock onto the magnetosphere was at 06:14 UT (32 minutes after DSCOVR observations), corresponding, under the assumption of a planar propagation, to the morning side of the magnetopause (7:01 LT - Figure 2g).

As a consequence of the IP shock impact, the magnetospheric field lines configuration reveal a large magnetopause com-

pression from 10 $R_E$ to 7.1 $R_E$ as both predicted by the TS04 model and observed by GOES14/GOES15 satellites. At the arrival of the ICME the magnetosphere is stretched and twisted as a consequence of the action of the magnetopause and the ring current alone between August 25, 2018 at 13:55 UT and August 26, 2018 at 8:15 UT (corresponding to the main phase of the geomagnetic storm, at ground), and of the concurring contribution of both the ring and the tail currents between August 26, 2018 at 8:15 UT and August 31, 2018 (corresponding to the recovery phase of the geomagnetic storm, at ground). This

scenario is confirmed by the simulation of a modified TS04 model set with the previous magnetospheric current assumptions, which well represents the behaviour of the observations at geosynchronous orbit (red dashed lines in Figure 4). A similar situation is obtained at LEO orbit on CSES satellite (figure 5), where the magnetospheric origin field variations (low frequency contributions) are induced by the action of both the symmetric part of the ring current and tail current along $B_{N,LF}$ and of the asymmetric part of the ring current along $B_{E,LF}$ (*Piersanti et al.*, 2017b), as confirmed by the TS04* model previsions.

Differently from GOES observations, CSES shows also variations at higher frequencies ($\sim 0.025$ mHz $<$ f $<\sim 3$ mHz), which are of both ionospheric current systems and the magnetospheric-ionospheric coupling origin contributions. Our interpretation of the huge positive then negative variations observed during the main phase along both the horizontal components, is due to the loading-unloading process between the magnetosphere and the ionosphere (*Consolini and De Michelis*, 2005; *Piersanti et al.*, 2017b). On the other hand, the variations observed during the recovery phase are due to the ionospheric DP-2 current

system (*Villante and Piersanti*, 2011; *Piersanti and Villante*, 2016; *Piersanti et al.*, 2017b).

At ground, during the main phase, the disturbance fields observed at latitudes between $50°$ and $75°$, on the night side, are due to the intensification of the polar westward electrojet. In addition, on August 26, 2018, the pattern of the polar electrojects are consistent with a ICME impacting on the morning side of the magnetosphere. In fact, as expected (*Wang et al.*, 2010; *Piersanti and Villante*, 2016; *Pilipenko et al.*, 2018), the greater disturbance for both the westward and eastward electrojects are located

around 7:00 LT, (central panels of Figure 9). In the same day, the injection of energetic particles from the magnetotail in the equatorial plane increased the ring current, generating at lower latitudes strong depression of the horizontal field magnitude on the Earth's surface (right panels in Figure 9). During the recovery phase, we observed a return of the horizontal component of the geomagnetic field to pre-storm values due to the decrease the ring current amplitude(*Piersanti et al.*, 2017b).

From an ionospheric point of view, to figure out whether the significant increase of electron density irregularities recorded in

terms of RODI, especially during the main phase, affected navigation systems, we estimated loss of lock from the vTEC Swarm data. No loss of lock has been found, which means that the event was weak in terms of space weather effects on navigation systems. This fact is supported by Figure 7 showing that loss of lock occurs mainly for really high values of ROTI, values never recorded during the period under analysis.

The amplitude of the geomagnetically induced currents index (*Marshall et al.*, 2011; *Tozzi et al.*, 2019), evaluated during

the August 2018 geomagnetic storm, reached very high values above $60°$ N of geomagnetic latitude. A direct comparison to





St. Patrick event showed that although the different storm intensities, the GIC hazard was extreme during the August 2018 event, while "only" high in the March 2015 event. On the other hand, both storms present very low values of GIC-index at low-mid latitudes, suggesting a greater participation of the ring current system. In any case, it is possible to observe the different impact of this storm at two different MLTs that is in good agreement with the reconstruction of the geomagnetic disturbance
as recorded on the ground (see Figure 9).

## 6 Conclusions

The solar event occurred on August 20, 2018 has been capable to increase the intensity of the various electric current systems flowing in the magnetosphere and ionosphere activating a chain of processes which cover a wide range of time and spatial scales and, at the same time, to activate strong interactions between various regions within the solar-terrestrial system. The
geomagnetic storm and the magnetospheric substorms occurred in the days following the solar event are the typical signatures of this chain of processes. The long lasting reconnection at the dayside magnetopause led to an increase of magnetospheric circulation, to an injection of particles into the inner magnetosphere and more in general provided free energy which was stored in the magnetosphere and leaded to a worldwide magnetic disturbance. The development of such disturbance has led to an increase of currents in the ionosphere accompanied by the auroral activity and by a shift equatorward of the auroral
electrojets and to the growth of the ring current (i.e. the westward toroidal electric current flowing around the Earth on the equatorial plane) accompanied by a worldwide reduction of the horizontal components of the geomagnetic field at low- and mid-latitudes. Rapid geomagnetic variations induced geoelectric fields on the conducting ground responsible for GICs whose intensity, as expected, varied with geomagnetic latitude (*Tozzi et al.*, 2018, and references therein). The amplitude of these currents, quantified by means of the GIC index has reached values corresponding to "high" and "extreme" risk levels above $60°$
N of geomagnetic latitude. However, no failures or malfunctioning are reported in literature. A higher sampling of the different geomagnetic latitudes would have allowed to more precisely depict GIC variations with latitude.

This storm is one of the few strong geomagnetic storms (G3-class, https://spaceweather.com/) that occurred during the current, $24^{th}$, solar cycle and represents one of those cases which have clearly shown how unpredictable space weather is and how much work is needed to make reliable predictions of the effects that solar events could have on the terrestrial environment.
Indeed, the CME emitted by the Sun in the days before the occurrence of the geomagnetic storm showed no features that would suggest the occurrence of important effects in the circumterrestrial environment or at ground. Indeed, as numerous studied have shown, the magnitude and features of geomagnetic storms depend not only on solar wind plasma parameters and on the values of the IMF, but also on their evolution (*Piersanti et al.*, 2017b, and references therein). Failing to predict the intensity of the August 26, 2018 storm has meant not being able to estimate correctly its effects on anthropic systems such as satellites,
telecommunications, power transmission lines and the safety of airline passengers. This confirms that, despite considerable advances in comprehending the drivers of space weather events, there is still room for improvements of their forecasting. It is important to underline that the future capabilities of forecasting if, where and when an event occurs and how intense it will be,





will depend on our understanding of the physical processes behind the dynamics in the near-Earth space (*Singer et al.*, 1998; *Pulkkinen*, 2015; *Piersanti and Carter*, 2019).

As a closing remark, we stress that, from a space weather point of view, this kind of comprehensive analysis plays a key role to better understand the complexity of the processes occurring in the Sun-Earth system that determines the geoeffectiveness of solar activity manifestations.

**Appendix A: RODI calculation**

To define RODI it is necessary to calculate the rate of change of the electron density (ROD), defined as:

$$\mathrm{ROD}(t) = \frac{N\mathrm{e}(t+\delta t) - N\mathrm{e}(t)}{\delta t}, \tag{A1}$$

where $N\mathrm{e}(t)$ and $N\mathrm{e}(t+\delta t)$ are the electron density measured by the Langmuir Probe on board the CSES satellite at time $t$ and $(t+\delta t)$, respectively; $\delta t = 3$ s since the CSES Langmuir Probe sampling rate is 1/3 Hz. Electron density values are provided in the form of continuous time series as a function of time; however, missing measurements are possible, an issue that has to be taken into account from a computational point of view. Consequently, time and electron density measured values are

indexed through an index $k$ running on the whole time series. With this approach, the $k^{\mathrm{th}}$ ROD value is calculated as:

$$\mathrm{ROD}_k = \frac{N\mathrm{e}_{k+1} - N\mathrm{e}_k}{t_{k+1} - t_k}, \tag{A2}$$

where $N\mathrm{e}_k$ is the electron density measured at a specific time $t_k$ and $N\mathrm{e}_{k+1}$ is the electron density measured at time $t_{k+1}$, only when the condition $(t_{k+1} - t_k) = \delta t = 3$ s is satisfied, i.e. for time consecutive measurements (according to the Langmuir Probe sampling rate). RODI is the standard deviation of ROD values in a running window of $\Delta t$. Specifically, to calculate

RODI, only ROD values calculated between $(t - \frac{\Delta t}{2})$ and $(t + \frac{\Delta t}{2})$ are taken into account. Then, RODI at each definite time $t$ is:

$$\mathrm{ROD}(t) = \sqrt{\frac{1}{N-1} \sum_{t_i = t - \frac{\Delta t}{2}}^{t + \frac{\Delta t}{2}} \left| \mathrm{ROD}(t_i) - \overline{\mathrm{ROD}}(t) \right|^2}; \tag{A3}$$

$\mathrm{ROD}(t_i)$ are ROD values falling inside the window centered at time $t$ and $\Delta t = 30$ s wide. $N$ is the number of ROD values in the window, while $\overline{\mathrm{ROD}}(t)$ is the corresponding mean, that is:

$$\overline{\mathrm{ROD}}(t) = \frac{1}{N} \sum_{t_i = t - \frac{\Delta t}{2}}^{t + \frac{\Delta t}{2}} \mathrm{ROD}(t_i). \tag{A4}$$



From a computational point of view the $k^{\text{th}}$ RODI value is calculated as:

$$\text{RODI}_k = \sqrt{\frac{1}{N-1} \sum_{i=-j}^{j} \left| \text{ROD}_{k+i} - \overline{\text{ROD}}_k \right|^2}, \tag{A5}$$

where $\text{ROD}_k + i$ are ROD values falling inside the window of width $(2j + 1)$, with $j = 5$, centered at the index $k$. To take into account possible missing measurements in the time series, only ROD values satisfying the condition $|t_{k+i} - t_k| \leq \frac{\Delta t}{2} = 15$

s are considered. $N$ is the number of ROD values (at most 11) falling in the window, and $\overline{\text{ROD}}_k$ is the corresponding mean of these $N$ values, that is:

$$\overline{\text{ROD}}_k = \frac{1}{N} \sum_{i=-j}^{j} \text{ROD}_{k+i}. \tag{A6}$$

Finally, RODI is calculated only when at least 6 ROD values fall in the window (the half plus one of maximum values inside a window, with $\delta t = 3$ s and $\Delta t = 30$ s). In this way, windows poorly populated, and consequently not statistically reliable, are

discarded.

*Author contributions.* MP managed the manuscript, made the analysis the magnetic field data from both satellite and ground observations, and concurred to the discussion of the results; PDM analyzed the geomagnetic data and concurred to the discussion of the results; RT made the GIC analysis and concurred to the discussion of the results; DDM analyzed Solar data and run the simulation of the ICME propagation; MP and AP analyzed the ionospheric plasma data and evaluated both ROTI and RODI; GC and VQ analyzed the magnetospheric field data

and concurred to the discussion of the results; SDM analyzed the Solar Wind data; PD validate and processed the CSES data; ML made the interplanetary analysis; MFM made the magnetospheric analysis; all the authors approved the final version of the manuscript.

*Competing interests.* The Author's declare that no competing interest are present.

*Acknowledgements.* The results presented in this paper rely on data collected at magnetic observatories. SDO Data is courtesy of NASA SDO/AIA and the HMI science teams. SOHO Data supplied courtesy of the SOHO/MDI and SOHO/EIT consortia. SOHO is a project of

international cooperation between ESA and NASA. This research has made use of data provided by the Heliophysics Event Knowledgebase. DSCOVR data were obtained from the NOAA's National Centers for Environmental Information (NCEI) Data Center. We thank the national institutes that support them and INTERMAGNET for promoting high standards of magnetic observatory practice (www.intermagnet.org). This work made use of the data from CSES mission (http://www.leos.ac.cn/), a project funded by China National Space Administration and China Earthquake Administration in collaboration with Italian Space Agency and Istituto Nazionale di Fisica Nucleare. The authors kindly

acknowledge N. Papitashvili and J. King at the National Space Science Data Center of the Goddard Space Flight Center for the use permission





of 1-minute OMNI data and the NASA CDAWeb team for making these data available. We acknowledge use of NOAA Space Weather Prediction Center for obtaining GOES magnetometer data. The European Space Agency (ESA) is acknowledged for providing the Swarm data. The official Swarm website is http://earth.esa.int/swarm. M. Piersanti thanks the Italian Space Agency for the financial support under the contract ASI "LIMADOU scienza" n° 2016-16-H0. This research work is supported by the Italian MIUR-PRIN on *Circumterrestrial*

*environment: impact of Sun - Earth interaction*.



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

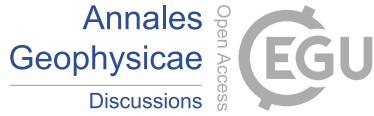

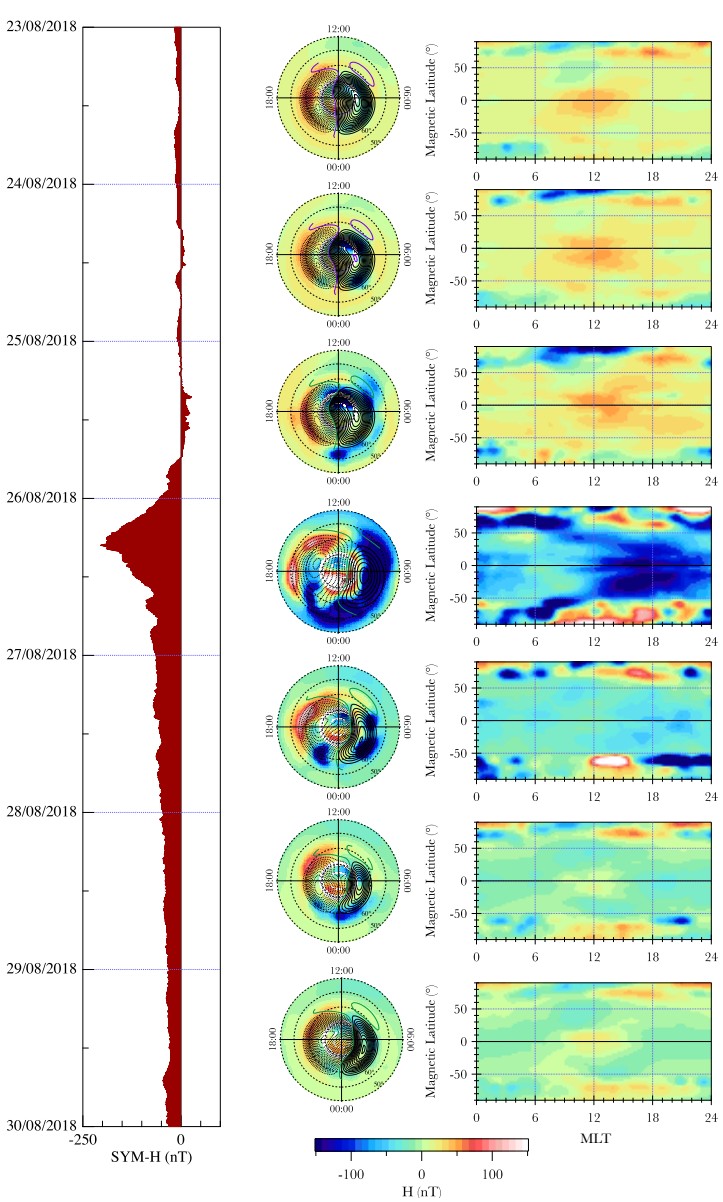

**Figure 9.** On the left the evolution of the SYM-H index; in the middle column, daily polar view maps of the horizontal field magnitude in the Northern Hemisphere. The convection patterns derived from the SuperDARN based model of (*Thomas and Shepherd*, 2018) are over plotted on the horizontal field magnitude; on the right column the worldwide view of the same magnetic field component. Data are reported in geomagnetic latitude and MLT, referring to a period of seven days from August 23, 2018 to August 29, 2018.



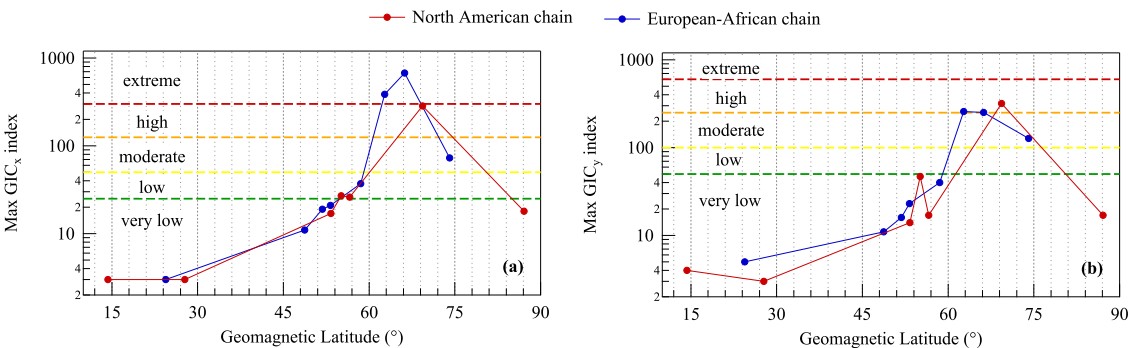

**Figure 10.** Maximum value of the GIC indices, occurred in the time interval from August 25, 2018 at 18:00 UT August 26, 2018 at 18:00 UT, as observed at the magnetic observatories of both the North American and European-African latitudinal chain. In detail, panel a) displays the maximum values of the $GIC_x$ index, panel b) the maximum values of the $GIC_y$ index. Colored dashed lines indicate the thresholds between the different risk levels as defined by *Marshall et al.* (2011).

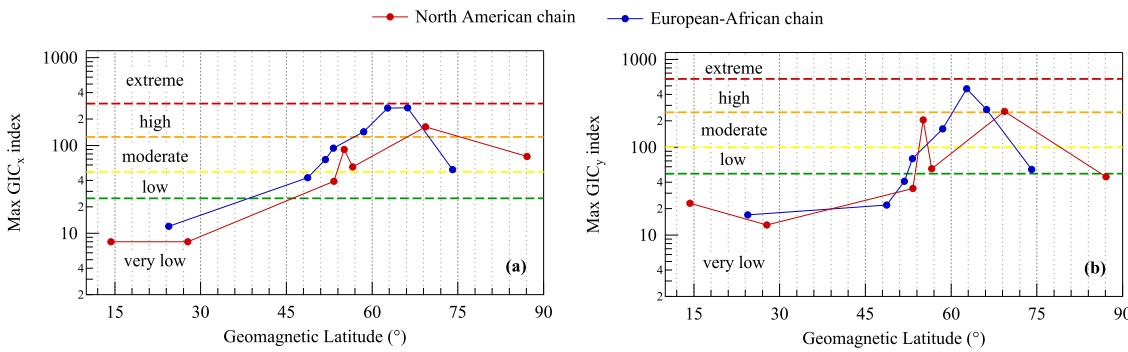

**Figure 11.** Maximum value of the GIC indices, occurred in the time interval from March 17, 2015 at 04:00 UT to March 18, 2015 at 04:00 UT, as observed at the magnetic observatories of both the North American and European-African latitudinal chain. In detail, panel a) displays the maximum values of the $GIC_x$ index, panel b) the maximum values of the $GIC_y$ index. Colored dashed lines indicate the thresholds between the different risk levels as defined by *Marshall et al.* (2011)