# Peer review of "From the Sun to the Earth: August 25, 2018 geomagnetic storm effects"

_Annales Geophysicae, 2019_

## Referee Comment (RC1) · Anonymous Referee #1 · 12 Feb 2020

Manuscript # angeo-2019-165

Title: From the Sun to the Earth: August 25, 2018 geomagnetic storm effects

Authors: Piersanti et al., 2020

General comments

This paper presents an end-to-end study of the geomagnetic storm on August 25, 2018. The analysis begins with an examination of the solar eruption to the effect in the magnetosphere-ionosphere coupled system and finally the ground effects. I find that the results are very interesting for the space weather science community and have direct practical consideration for providers/users of space weather services/information. Therefore, I recommend that the manuscript be excepted for publication after consid-

erable modifications listed below:

Specific comments

Page 3, Line 55-57: How is RODI different from ROTI (rate of change of TEC index)? Please explain for benefit of readers. After getting to the end of the paper, I find a detailed description of RODI computation. It would be worthwhile pointing out to the reader here that you have a detail description in the Appendix.

Page 4, Figure 1: I think it will be better to draw contours around the features than to use single post marks for the position.

Page 6, Figure 1: It's hard to see the Venus green triangle with the large green shaded area. I suggest changing the triangle to a different color.

Page 6, Figure one caption: There are two green areas 1 light and other darker. What does the dark green area represent?

Page 11, Figure 6: While RODI did recover on August 27, note that some high RODI values are still present in the Asia/Australia equatorial region. I think it is important to mention this feature in the text.

Page 11, Line 216: Why are you doing this? This may not be obvious to all readers; thus, it must be explained.

Page 12, Lines 242-244: My understanding is that IMTERMAGNET data is sampled at 1 second and filtered down to 1 minute to avoid aliasing effects. Can you comment on that?

Page 14, Figure 8: What is the longitudinal range for the green chain in North America? It seems quite spread out compared to the European-African chain.

Page 14, Lines 260-261: Please explain how the removal has been done. What baseline did you use for this process? Was a common baseline applied or did you do it separately for each station?

Page 17, Line 349-250: Did is occur in the same local time zone? What about consideration of seasonal effects?

Technical corrections

Page 3, Line 57: Geomagnetically Induced Current (GIC).

Page 3, Line 58: Change Geomagnetically Induced Current (GIC) to "GIC".

Page 3, Line 71: Change "FoV's" to Field of View (FoV).

Page 3, Lines 72-73: Delete (Field of View).

Page 5, Lines 106-107: Did you mean increase from 11 - 30 cm-3?

Page 5, Line 114: Change "overtook" to "overtaken".

Page 7, Line 134: Add space between words "Space weather." and "Its".

Page 7, Line 158: Add space between words "model and "(Tsyganenko".

Page 12, Lines 247-248: Delete one ". . . and the SuperDARN observations as well, . . .".

Page 15, Line 296: Change ". . . on turn . . ." to "in turn".

Page 16, Line 300: ". . . makes of GIC a DC current that flows into . . ." change to ". . . makes the GIC a quasi-DC current compared to the . . ."

---

## Referee Comment (RC2) · Anonymous Referee #2 · 13 Feb 2020

Review report on the manuscript "From the Sun to the Earth: August 25, 2018 ge-omagnetic storm effects", submitted to Annales Geophysicae by Piersanti et al. for consideration of publication.

Manuscript summary

This manuscript deals with the effects caused by a geomagnetic storm associated with solar perturbations (CME, CIR, HSS) on the geospace environment and on the ground in the period 20-27 August 2018. The authors tracked a CME detected on the Sun and estimated its speed and travel time to 1 AU. The authors then used multi-spacecraft and multi-instrument analyses to study the chain reaction of geospace currents to these drivers and their effects on modern technological systems. The authors also looked at GPS data to evaluate the impact of the storm on these very important systems.

[Figure]

Specifically, the authors looked at geosynchronous, LEO, and ground magnetic field data. Additionally, the authors estimated coefficient/indices for GICs associated with this storm. By comparing these effects to the well-known 2015 St. Patrick's Day storm, which was slightly more intense than the August 2018 storm, the authors concluded that (i) effects on GPS were more severe in 2015; but (ii) effects on GICs were more severe in 2018. These were perhaps the most important results found by the authors at least from a space weather-related standpoint. As a result, intensities of geomagnetic storms can't be used as a standalone parameter to predict and forecast space weather effects on technological infrastructure.

I think the paper is well-written and chronologically organized; this type of investigation, i.e., analyses of chain response characterized by multi-instrument investigations, are very important to advance our knowledge of space weather phenomena. However, I think the paper can be improved if the authors take in consideration my general and specific comments listed below.

General comments

1. The paper can be significantly improved if the authors look more carefully into the driver effects. For example, the authors mention the CME observed on 20 August was a very slow, and consequently weak, CME. Therefore, this kind of CME would drive a very weak IP shock in its leading edge, if any, while traveling in the heliosphere. However, given the IMF and solar wind data detected at ∼L1 presented by the authors in Figure 2, I can't tell there is a fast forward interplanetary shock there. There is no clear positive jumps in solar wind parameters (T, Np, V) and IMF (B). A positive sudden impulse is clearly and usually seen in ground magnetometer data in response to an IP shock impact. However, I could not see a clear SI event when I plotted SYM-H here on my side, neither can I see a clear SI signature in SYM-H data shown in the left column of figure 9. Please clarify this: where is the shock around L1? If there is one, does it clearly satisfy the Rankine-Hugoniot conditions? This is very important because this would impact your discussion of inward magnetopause motion (from 10 to 7 Re,

approximately). As it it I think it was because of magnetopause erosion caused by the gradual depletion of IMF Bz, which is not shown in the manuscript.

2.   Now, still looking at Figure 2, there is something I think ins't clear: What do the edges of the light red highlighted area indicate?   The IP shock onset or the front boundary of the magnetic cloud?   As a matter of fact, according to the CME list provided by Ian Richardson (http://www.srl.caltech.edu/ACE/ASC/DATA/level3/icmetable2.htm#(a)), the CME of 25 August 2018 detected at L1 did not have a magnetic cloud associated with it. Can you clearly say why your CME had a magnetic cloud, even though it was a weak CME and most likely did not drive a shock in its front edge at 1 AU?

3. According to Figure 2 and Figure 3, there was a CIR/HSS right after the CME. If the geomagnetic storm of 25-26 August 2018 occurred as a result of the impacts of a CME and a CIR back-to-back, why possible effects of these combined rivers are not discussed in the text? By looking at the SYM-H plot (Figure 9), one can speculate that the magnetosphere was starting to recover from the CME when the CIR arrived. Does this have any impacts on the subsequent geomagnetic activity? Do you think SYM-H effects were amplified with the CIR arrival?  This might explain why the GIC effects in 2018 storm were stronger than the effects in the 2015 storm.  However, it is very common to see a clear SI signature preceding a strong geomagnetic storm, which did not occur on 25 August 2018. In other words, does the CIR play any roles in increasing the storm effects discussed in the manuscript?

Specific comments

1. Line 12. Please explicitly state that this is the 2015 St. Patrick's Day storm.

2. Lines 17-18. The way this sentence is written, one can wrongly understand that geomagnetic storms are only caused by CMEs. Please re-write this sentence to eliminate this inaccurate statement.

3. Line 18: The reference Piersanti et al. (2017b) doesn't appear in the reference list. I didn't check all references; please do so.

4. Line 27. Change "the Sun" to "solar".

5. Lines 27-28. Sun energy directly deposited in the polar ionosphere. Do you mean EUV radiation? If so, EUV radiation is directly deposited by the Sun in all dayside latitudes. I think what you mean is that solar radiation increases dayside conductivity which in turn facilitates the flow of electric current in the ionosphere? Please clarify this. It looks awkward to read.

6. Line 37. Hapgood (2019) discusses GIC effects of the May 1921 superstorm that were associated with fires in New York City. Therefore, space weather related effects can be dangerous to human life. If the authors are interested, here is the reference:

Hapgood, M. (2019), The Great Storm of May 1921: An Exemplar of a Dangerous Space Weather Event, Space Weather, 17 (7), 950–975, doi:10.1029/2019SW002195.

7. Line 41. Please state what defines a G3 geomagnetic storm.

8. Line 63. Please give a number for the reader to have an idea of how fast a slow CME goes.

9. Line 89. Add "dynamic" before "pressure".

10. Line 114. Change "overtook" to "overtaken". Same in line 360.

11. Figure 3. It is hard to see Venus as represented by the green triangle.

12. Line 126. "Possibly" reads better than ""probably.

13. Line 156. Should "IP2" be "IPS"?

14. Line 215. Please explicitly state of loss of lock on GPS satellites means. Add a reference if appropriate.

15. Line 219. Please state that storm intensity is represented by Dst/SYM-H data.

16. Line 246: Remove a "the" (end of line).

17. Line 254. The traditional reference for the SYM-H index is Iyemori (1990),

Iyemori, T. (1990), Storm-time magnetospheric currents inferred from mid–latitude geomagnetic field variations, Journal of Geomagnetism and Geoelectricity, 42(11), 1249–1265, doi:10.5636/jgg.42.1249.

18. Line 260. Do you mean you are removing the background magnetic field computed by the IGRF model? Please clarify.

19. Line 274. "on August" should be "On August".

20. Line 297. "on turn" should be "in turn".

21. Line 312. Please include a table with the stations' names and abbreviations and refer to it instead of referring to the IAGA website.

22. Line 346. Change "has been" too "was". No continuity here.

23. Line 413. Change "leaded" to "led".

24. Line 431. Change "comprehending" to "understanding".

25. My apologies, but I read in a few places mentions to "polar electrojets". Do you mean auroral electrojets? Usually, these electric currents have their effects expressed by the AU, AL, and AE indices. If so, please clarify and change it accordingly. Additionally, it would be interesting to plot these indices in another column in Figure 9.

---

## Author Comment (AC1) · 29 Feb 2020

We thank the Reviewer who appears to agree with the significance of our results and comments our work as suitable for publication after minor revisions. In the revised version all her/his suggestions have been considered, namely:

Specific Comments:

Page 3, Line 55-57: How is RODI different from ROTI (rate of change of TEC index)? Please explain for benefit of readers. After getting to the end of the paper, I find a detailed description of RODI computation. It would be worthwhile pointing out to the reader here that you have a detail description in the Appendix. According to the re-mark made by the reviewer the following text has been added in the Introduction: "To

characterize ionospheric irregularities and fluctuations, we used the Rate Of change of electron Density Index (RODI; specifications about the calculation of this index can be found in the Appendix A) estimated from the electron density measured by CSES. To understand how the presence of such irregularities could have affected navigational systems, we have also considered total electron content (TEC) values from Swarm to highlight possible loss of lock, condition under which a Global Positioning System (GPS) receiver no longer tracks the signal sent by the satellite with a consequent degradation of the positioning accuracy (Jin and Oksavik, 2018; Xiong et al., 2018)." RODI is an index that can be calculated only along the orbit of the satellite, because it is based on the electron density measurements (which are punctual) made by the satellite; this is why for each definite moment of time it is possible to calculate only one value of RODI. ROTI is instead based on TEC values (which is the integral of the electron density along the direction satellite-receiver) calculated by the GPS receiver for each satellite in view so that, unlike RODI, it is possible to obtain several values of ROTI for each definite moment of time. Both indexes characterize similarly the ionosphere in terms of irregularities; anyhow, the added value of ROTI is that it can highlights also possible loss of lock, as it is well visible in the bottom panel of Figure 7.

Page 4, Figure 1: I think it will be better to draw contours around the features than to use single post marks for the position. We changed the posts into coloured contours. Page 6, Figure 3: It's hard to see the Venus green triangle with the large green shaded area. I suggest changing the triangle to a different colour. The colour of the fast solar wind stream has been changed to grey to solve both problems and increase the readability of the image. Page 6, Figure three caption: There are two green areas 1 light and other darker. What does the dark green area represent? The colour of the fast solar wind stream has been changed to grey to solve both problems and increase the readability of the image.

Page 11, Figure 6: While RODI did recover on August 27, note that some high RODI values are still present in the Asia/Australia equatorial region. I think it is important to

mention this feature in the text. According to the remark made by the reviewer the text was revised as:

"Significant high values of RODI, spreading all over the meridian during the main phase of the storm (August 25 and 26,2018, especially the latter), for both nighttime and daytime, are clearly seen, while on August 27, 2018, the RODI index comes back to lower values, even though some significant values of RODI are still visible in the Asian-Australian longitude sector at equatorial latitudes."

Page 11, Line 216: Why are you doing this? This may not be obvious to all readers; thus, it must be explained. According to the remark made by the reviewer the following text has been added at page XXXXX: "As recommended in the Swarm L2 TEC product description (available at https://earth.esa.int/documents/10174/1514862/Swarm\_Level-2\_TEC\_Product\_Description) only TEC data with corresponding elevation angles $\geq 50°$ have been taken into account, because considered more reliable."

Page 12, Lines 242-244: My understanding is that IMTERMAGNET data is sampled at 1 second and filtered down to 1 minute to avoid aliasing effects. Can you comment on that? We changed the sentence about INTERMAGNET data used in this study and clarified that, although INTERMAGNET provides also 1-second geomagnetic data, we analysed 1 minute data since to the purpose of mapping of the daily averaged disturbance the 1-minute resolution was enough.

Page 14, Figure 8: What is the longitudinal range for the green chain in North America? It seems quite spread out compared to the European-African chain. The spread in longitude of the two latitudinal chains seem different since the map shown in Figure 8 is in geographic coordinates (this is now specified in the caption). However, the observatories belonging to the two chains have been chosen to have of geomagnetic longitudes that are spread over a similar range, this range is around 40° (specifically 42.6° for the Noth-American chain and 42.3° for the European-African chain). To clarify

this point to the readers we have changed the sentence "...geomagnetic longitudes in a range of about ±20° around a central longitude." into "that are spread over a range of ïĆż 40° around a central longitude".

Page 14, Lines 260-261: Please explain how the removal has been done. What baseline did you use for this process? Was a common baseline applied or did you do it separately for each station? We agree with the reviewer. We added a sentence about the baseline removal process we used for our analysis. Namely, for each ground station, we used the CHAOS-6 model to remove both the internal and crustal origin field from the magnetic data. So, we are confident that the residual magnetic field is of external origin (ionosphere + magnetosphere).

Page 17, Line 349-350: Did it occur in the same local time zone? What about consideration of seasonal effects? We thanks the reviewer for his/her suggestions. We checked about possible seasonal effects explaining the differences about GIC-index between 2015 St. Patrick day storm and 2018 August storm. In terms of GIC index, the maximum effect of the 2015 St. Patrick storm occurred on the dayside for both chains. We observe that while for the 2018 August storm the ICME impacted the magnetopause approximately on the morning, for the 2015 St Patrick's storm the impact occurred practically on the nose of the magnetopause. We believe that this could, at least partly, explain the differences in the behaviour of the two storms in terms of GIC index. We have integrated the manuscript with the information above. To discuss seasonal effects we should take into consideration many more geomagnetic storms, this is beyond the purpose of the present manuscript.

Technical corrections:

We made all the technical and grammar corrections proposed by the reviewer.

---

## Author Comment (AC2) · 29 Feb 2020

We thank the Reviewer who appears to agree with the significance of our results and comments our work as suitable for publication after minor revisions. In the revised version all her/his suggestions have been considered, namely: General comments

1. The paper can be signiïfįcantly improved if the authors look more carefully into the driver effects. For example, the authors mention the CME observed on 20 August was a very slow, and consequently weak, CME. Therefore, this kind of CME would drive a very weak IP shock in its leading edge, if any, while traveling in the heliosphere. However, given the IMF and solar wind data detected atâĹijL1 presented by the authors in Figure 2, I can't tell there is a fast forward interplanetary shock there. There is no

clear positive jumps in solar wind parameters (T, Np, V) and IMF (B). A positive sudden impulse is clearly and usually seen in ground magnetometer data in response to an IP shock impact. However, I could not see a clear SI event when I plotted SYM-H here on my side, neither can I see a clear SI signature in SYM-H data shown in the left column of figure 9. Please clarify this: where is the shock around L1? If there is one, does it clearly satisfy the Rankine-Hugoniot conditions? This is very important because this would impact your discussion of inward magnetopause motion (from 10 to 7 Re, approximately). As it is I think it was because of magnetopause erosion caused by the gradual depletion of IMF Bz, which is not shown in the manuscript. We thank the reviewer for his/her useful comments and we completely agree you. There is no clear evidence of an IP shock preceding the magnetic cloud, despite the presence of a positive SI at midlatitude ground stations. This is probably due to the impact of the front boundary of the magnetic cloud coupled with a southward switching of the Bz,IMF. Thus, we change accordingly the discussion about GOES and L1 satellite data.

2. Now, still looking at Figure 2, there is something I think ins't clear: What do the edges of the light red highlighted area indicate? The IP shock onset or the front boundary of the magnetic cloud? As a matter of fact, according to the CME list provided by Ian Richardson (http://www.srl.caltech.edu/ACE/ASC/DATA/level3/icmetable2.htm#(a)), the CME of 25 August 2018 detected at L1 did not have a magnetic cloud associated with it. Can you clearly say why your CME had a magnetic cloud, even though it was a weak CME and most likely did not drive a shock in its front edge at 1 AU? The red area of Figure 2 indicates the front boundary of the magnetic cloud. In our opinion the CME of August 25, 2018 has a magnetic cloud. In fact, following Burlaga et al. (1981), we clearly found at L1 point a region of enhanced magnetic field strength, smooth rotation of the magnetic field vector, and low proton temperature. Since the ICME under analysis was very slow and weak, it is not characterized by its typical structure (i.e. a fast-mode shock wave followed by a dense (and hot) sheath of plasma (the downstream region of the shock) and a magnetic cloud), but it contains only the magnetic cloud. In addition,

as stated by Lepping, R. P. et al. (1990) "a magnetic cloud presents a typical speed of 450 km/s and magnetic field strength of 20 nT", which are consistent with our satellites observations.

3. According to Figure 2 and Figure 3, there was a CIR/HSS right after the CME. If the geomagnetic storm of 25-26 August 2018 occurred as a result of the impacts of a CME and a CIR back-to-back, why possible effects of these combined rivers are not discussed in the text? By looking at the SYM-H plot (Figure 9), one can speculate that the magnetosphere was starting to recover from the CME when the CIR arrived. Does this have any impacts on the subsequent geomagnetic activity? Do you think SYM-H effects were amplified with the CIR arrival? This might explain why the GIC effects in 2018 storm were stronger than the effects in the 2015 storm. However, it is very common to see a clear SI signature preceding a strong geomagnetic storm, which did not occur on 25 August 2018. In other words, does the CIR play any roles in increasing the storm effects discussed in the manuscript? Thank you very much for your observations. Looking at the "quicklook" of temporal trend of the AE indices (wdc.kugi.kyoto-u.ac.jp/aedir/), we have noticed that during the recovery phase, the AL index is characterized by strong decreases caused by sudden variations of the Bz,IMF carried by the CIR ("back-to-back" to the ICME), which correspond to an increase of the westward auroral electrojet current. Such effect is clearly visible in the decrease of the magnetic field recorded on the ground (Figure 9 of the manuscript) in the down side. We discussed this point in the manuscript. Anyway, in our opinion, this effect cannot be visible in the GIC plot, because it was evaluated at the time corresponding to the minimum Sym-H value, when the CIR have not yet impact onto the magnetosphere.

Specific comments

1. Line 12. Please explicitly state that this is the 2015 St. Patrick's Day storm. Done

2. Lines 17-18. The way this sentence is written, one can wrongly understand that geomagnetic storms are only caused by CMEs. Please re-write this sentence to eliminate

this inaccurate statement. Done

3. Line 18: The reference Piersanti et al. (2017b) doesn't appear in the reference list. I didn't check all references; please do so. Checked

4. Line 27. Change "the Sun" to "solar". Done

5. Lines 27-28. Sun energy directly deposited in the polar ionosphere. Do you mean EUV radiation? If so, EUV radiation is directly deposited by the Sun in all dayside latitudes. I think what you mean is that solar radiation increases dayside conductivity which in turn facilitates the flow of electric current in the ionosphere? Please clarify this. It looks awkward to read. We agree with the reviewer on the importance of EUV radiation in increasing the conductivity of the ionosphere, but we wanted to describe a different process due to plasma and not to radiation. For this reason we modified the sentence and clarified this point.

6. Line 37. Hapgood (2019) discusses GIC effects of the May 1921 superstorm that were associated with fires in New York City. Therefore, space weather related effects can be dangerous to human life. If the authors are interested, here is the reference: Hapgood, M. (2019), The Great Storm of May 1921: An Exemplar of a Dangerous Space Weather Event, Space Weather, 17(7), 950–975, doi:10.1029/2019SW002195. Reference added

7. Line 41. Please state what defines a G3 geomagnetic storm. We added the definition of a G3 storm using the Kp-index.

8. Line 63. Please give a number for the reader to have an idea of how fast a slow CME goes. We added a sentence clarifying this point.

9. Line 89. Add "dynamic" before "pressure". Done

10. Line 114. Change "overtook" to "overtaken". Same in line 360. Done

11. Figure 3. It is hard to see Venus as represented by the green triangle. The colour

of the fast solar wind stream has been changed to grey to solve both problems and increase the readability of the image.

12. Line 126. "Possibly" reads better than ""probably. Done

13. Line 156. Should "IP2" be "IPS"? Yes, we change it

14. Line 215. Please explicitly state of loss of lock on GPS satellites means. Add a reference if appropriate.

According to the remark made by the reviewer the following text has been added in the Introduction: "To characterize ionospheric irregularities and fluctuations, we used the Rate Of change of electron Density Index (RODI; specifications about the calculation of this index can be found in the Appendix A) estimated from the electron density measured by CSES. To understand how the presence of such irregularities could have affected navigational systems, we have also considered total electron content (TEC) values from Swarm to highlight possible loss of lock, condition under which a Global Positioning System (GPS) receiver no longer tracks the signal sent by the satellite with a consequent degradation of the positioning accuracy (Jin and Oksavik, 2018; Xiong et al., 2018)."

Jin, Y. and K. Oksavik, (2018), GPS scintillations and losses of signal lock at high latitudes during the 2015 St. Patrick's Day storm, J. 565 Geophys. Res., 123, https://doi.org/10.1029/2018JA025933. Xiong C., C. Stolle, and J. Park, (2018), Climatology of GPS signal loss observed by Swarm satellites, Annales Geophysicae 36, 679, https://doi.org/10.5194/angeo-36-679-2018.

15. Line 219. Please state that storm intensity is represented by Dst/SYM-H data. We agree with the reviewer. We added the Dst minimum value of the 2015 Saint Patrick Storm.

16. Line 246: Remove a "the" (end of line). Done

17. Line 254. The traditional reference for the SYM-H index is Iyemori (1990),

Iyemori, T. (1990), Storm-time magnetospheric currents inferred from mid–latitude geomagneticfieldvariations,JournalofGeomagnetismandGeoelectricity,42(11),1249– 1265, doi:10.5636/jgg.42.1249. We agree with the reviewer. We added the reference about Sym-H.

18. Line260. Do you mean you are removing the background magnetic field computed by the IGRF model? Please clarify. We agree with the reviewer. We added a sentence about the baseline removal process we used for our analysis. Namely, for each ground station, we used the CHAOS-6 model to remove both the internal and crustal origin field from the magnetic data. So, we are confident that the residual magnetic field is of external origin (ionosphere + magnetosphere).

19. Line 274. "on August" should be "On August". Done

20. Line 297. "on turn" should be "in turn". Done

21. Line 312. Please include a table with the stations' names and abbreviations and refer to it instead of referring to the IAGA website. According to Referee's suggestion we have prepared a table (Table 1 in the revised version of the manuscript) with the names and IAGA codes of the observatories of the two latitudinal chains. The table also provides their geomagnetic latitudes, longitudes and the difference in hour from the MLT of the $0°$ geomagnetic meridian at 0 UT.

22. Line 346. Change "has been" too "was". No continuity here. Done

23. Line 413. Change "leaded" to "led". Done

24. Line 431. Change "comprehending" to "understanding". Done

25. My apologies, but I read in a few places mentions to "polar electrojets". Do you mean auroral electrojets? Usually, these electric currents have their effects expressed by the AU, AL, and AE indices. If so, please clarify and change it accordingly. Additionally, it would be interesting to plot these indices in another column in Figure 9. We have changed "polar electrojets" in "auroral electrojects". Of course, it would be very

interesting to visualized the temporal trend of auroral electroject indices during the selected period. The problem is that, at the moment, these indices are not available for our period. Provisional data are available until March 2018.
* * *

---

## Author Response (AR2)

**Response to Anonymous Referee #1**

We thank the Reviewer who appears to agree with the significance of our results and comments our work as suitable for publication as it is.

**Response to Anonymous Referee #2**

We thank the Reviewer who appears to agree with the significance of our results and comments our work as suitable for publication after minor revisions.

In the revised version all her/his suggestions have been considered, namely:

1. **Comment #8 of my previous report was not addressed in the revised manuscript. Please do so. (Line 63. Please give a number for the reader to have an idea of how fast a slow CME goes).**

   An ICMEs is defined slow is $V_{ICME} - V_{SW} <= 0$ km/s, where $V_{ICME}$ is the ICME speed and $V_{SW}$ is the speed of the background solar wind [Iju (2013)]. We added this reference in the paper.

   Iju, T., Tokumaru, M. & Fujiki, K. Radial Speed Evolution of Interplanetary Coronal Mass Ejections During Solar Cycle 23. Sol Phys 288, 331–353 (2013). https://doi.org/10.1007/s11207-013-0297-5

2. **Line 76. Change "has been" too "was".**

   Done

3. **Line 85. Do you mean Lepping et al. (1995) and Stone et al. (1998)?**

   Yes. We changed accordingly.

4. **Line 86. Change "spacecrafts" too "spacecraft".**

   Done

5. **Lines 100-101. I think it would be better to state the shock ahead the ICME was dissipated while propagating in the interplanetary space, and consequently not observed at 1 AU, because the ICME as slow and weak. Another hypothesis that I can think of is that the CME hit Earth with one of its edges (Figure 2 and Figure 3). Therefore, the normal of that magnetic cloud structure was very inclined making it very hard for L1 solar wind monitors to detect a true shock (associated with the fact that the magnetic cloud was very weak). Please see review by Oliveira and Samsonov (2018) and references there in. This discussion could be interesting and useful. Oliveira, D. M., & Samsonov, A. A. (2018). Geoeffectiveness of interplanetary shocks controlled by impact angles: A review. Advances in Space Research, 61(1), 1-44. https://doi.org/10.1016/j.asr.2017.10.006.**

   We agree with the reviewer. We added a full discussion about this point and we added the reference you suggested.

**6. Figure 2. Shouldn't the units of theta and phi be in degrees? They are angles, aren't they?**

Yes, of course. We change it.

**7. Line 151. …which makes "it" very…**

Done

**8. Line 223. Do you mean minimum value of Dst?**

Yes, to make a rough comparison between two storms, typically the minimum values of both the Dst indices is taken into account. So, we reference here to the minimum Dst index reached during the 2015 St. Patrick day storm. Anyway, we added a sentence to specify that the value is the minimum.

**9. Lines 355-357. The authors still make a reference to the now debunked interplanetary shock of 24 August 2018. Do the authors mean the magnetic cloud structure? Please clarify. There are other places in the paper that should be addressed, for example, in the caption of Figure 2.**

We agree with reviewer. Concerning the "shock reference" between lines 355 and 357, we replaced it with "magnetic cloud structure" as you suggested. We double checked the manuscript in order to eliminate any other possible reference to the debunked interplanetary shock. Concerning the caption of Figure 2, here we referred to an IP shock that we detected into SW data but that we did not couple with the ICME structure. In this case, we prefer to report it into the data to give a complete vision of the SW data to a potential reader.

[revised manuscript text omitted]